# *Rickettsia felis* DNA recovered from a child who lived in southern Africa 2000 years ago

Riaan F. Rifkin [1,2✉], Surendra Vikram [1], Jaime Alcorta[3], Jean-Baptiste Ramond [1,2,3], Don A. Cowan [1], Mattias Jakobsson [4,5,6], Carina M. Schlebusch [4,5,6] & Marlize Lombard [5✉]

The Stone Age record of South Africa provides some of the earliest evidence for the biological and cultural origins of *Homo sapiens*. While there is extensive genomic evidence for the selection of polymorphisms in response to pathogen-pressure in sub-Saharan Africa, e.g., the sickle cell trait which provides protection against malaria, there is inadequate direct human genomic evidence for ancient human-pathogen infection in the region. Here, we analysed shotgun metagenome libraries derived from the sequencing of a Later Stone Age hunter-gatherer child who lived near Ballito Bay, South Africa, *c.* 2000 years ago. This resulted in the identification of ancient DNA sequence reads homologous to *Rickettsia felis*, the causative agent of typhus-like flea-borne rickettsioses, and the reconstruction of an ancient *R. felis* genome.

[1] Centre for Microbial Ecology and Genomics, Department of Biochemistry, Genetics and Microbiology, University of Pretoria, Hatfield, South Africa.
[2] Department of Anthropology and Geography, Human Origins and Palaeoenvironmental Research Group, Oxford Brookes University, Oxford, UK.
[3] Department of Molecular Genetics and Microbiology, Pontificia Universidad Católica de Chile, Santiago, Chile. [4] Department of Organismal Biology, Evolutionary Biology Centre, Uppsala University, Norbyvägen, Uppsala, Sweden. [5] Palaeo-Research Institute, University of Johannesburg, Auckland Park, South Africa. [6] SciLifeLab, Uppsala, Sweden. ✉email: riaanrifkin@gmail.com; mlombard@uj.ac.za

Southern Africa has long been a hotspot for research concerning the origins of *H. sapiens*[1]. The oldest genetic population divergence event of our species, at *c.* 350,000 to 260,000 years ago (kya), is represented by the genome of a hunter-gatherer child from Ballito Bay[2,3]. Fossil evidence exists for early *Homo sapiens* from ~259 kya[4], for late *H. sapiens* from at least 110 kya[5] and for cognitive-behavioural complexity since *c.* 100 kya[6–9]. Yet, despite the fact that pathogens have long exerted a significant influence on hominin longevity[10] and human genetic diversity[11], and given that diseases continue to shape our history[12], their influence on the biological and socio-cultural evolution of our species in Africa is routinely understudied. The gradual dispersal of *H. sapiens* from Africa into Asia and Europe was also accompanied by various commensal and pathogenic microbes[13–15]. The presence of specific TLR4 polymorphisms (i.e., pathogen-recognition receptors) in African, as well as in Basque and Indo-European populations, suggests that some mutations arose in Africa prior to the dispersal of *H. sapiens* to Eurasia[16]. In addition, the bio-geographic distribution of *Plasmodium falciparum*[17] and *Helicobacter pylori*[18] exhibits declining genetic diversity, with increasing distance from Africa mirroring past human expansions and migrations, with 'Out of Africa' estimates of ~58 kya and ~80 kya, respectively. Given the long association of *H. pylori* with humans, the current population structure of *H. pylori* has been regarded as mirroring past human expansions and migrations. From records such as these, it is evident that persistent exposure to pathogens exerted selective pressure on human immune-related genes[19–21], cognitive development[22] and social behaviour[23]. The potentially-adverse impact of diseases on ancient forager populations is exemplified by the fact that infectious, zoonotic, and parasitic diseases account for ~70% of deaths recorded amongst contemporary hunter-gatherer populations[24] (SI 2).

The DNA of a seven-year-old boy[2,3,25], who lived in South Africa near what is today the town of Ballito Bay *c.* 2000 years ago (ya), recently revised the temporal extent of our species by facilitating the re-calculation of the genetic time-depth for our species to between 350 to 260 kya[2] (Fig. 1). Here, we report on the molecular detection of a bacterial pathogen associated with this hunter-gatherer child (i.e., aDNA sample 'BBayA'). Originally excavated in the 1960s, the remains of the child have been dated by AMS radiocarbon (14C) to 1,980 ± 20 cal. BP (1936–1831 cal. BP at 95% probability) (Supplementary Information 1 (SI)). We were able to reconstruct an ancient genome for *Rickettsia felis*, a bacterium causing typhus-like flea-borne rickettsioses. Until now, *R. felis* has been widely viewed as a recent or emergent pathogen, first implicated as a cause of human illness in Texas, USA, in 1994[26,27]. On the contrary, our results show that *R. felis* was present by at least 2000 years ago amongst southern African Stone Age hunter-gatherers who did not practice animal husbandry or agriculture, and who did not follow a sedentary lifestyle.

## Results

**Identification of ancient pathogenic taxa.** Although there is substantial evidence for the selection of human genomic polymorphisms in response to pathogen-pressure in sub-Saharan Africa (SI 2), there is little direct evidence of ancient human-pathogen interactions in the region. To gain insight into the prehistoric incidence of human pathogens, we analysed eight shotgun metagenome libraries originating from the sequencing of the boy from Ballito Bay (Fig. 1). Initial taxonomic classification was achieved using Kraken2[28] (downloaded 30/10/2019) and a custom database of bacterial, archaeal, protozoal and viral genomes from the NCBI RefSeq database (https://www.ncbi.nlm.nih.gov/refseq/) (downloaded 14/03/2018). Supplementary Data 1 provides the taxonomic counts of raw reads derived

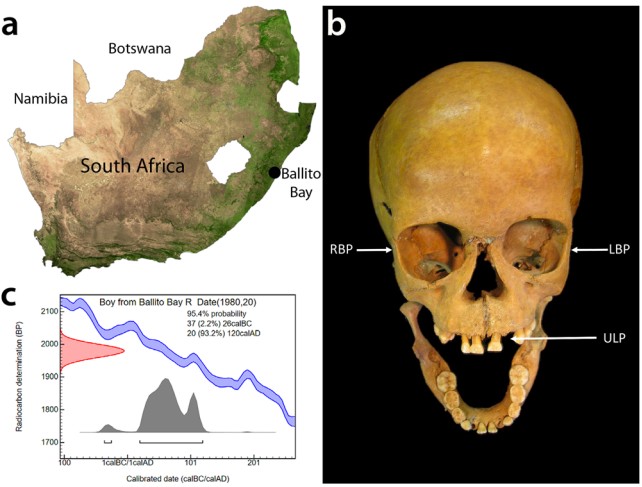

**Fig. 1 The provenience of the Later Stone Age hunter-gatherer skeletal remains recovered from a mound formed by a shell midden overlooking the beach in Ballito Bay, South Africa. a** The provenience of the Later Stone Age hunter-gatherer skeletal remains recovered from a mound formed by a shell midden overlooking the beach in Ballito Bay, KwaZulu-Natal Province, South Africa (map modified from GISGeography 'https://gisgeography.com/south-africa-map/'). **b** The cranial remains of the BBayA male child indicating aDNA sample sources, i.e., DNA was extracted and sequenced from bone samples acquired from the left petrous bone (*LPB*), right petrous bone (*RPB*) and the upper left premolar (*ULPM*) (image adapted from Pfeiffer et al., 2019). **c** The C14 date (1980 ± 20 cal. BP) obtained for the remains of the child.

from the analyses for the left petrous bone (LPB), right petrous bone (RPB), and the upper left premolar (ULPM) samples (Fig. 1). Pathogenic taxa were identified, and their reference genomes downloaded from the NCBI RefSeq database for downstream analysis. The mapping of candidate taxa was performed against bacterial and parasitic genomes, including two different *R. felis* assemblies, and a complete human genome, i.e., *H. sapiens* assembly GRCh38/hg39 (Supplementary Data 2) ("Methods"). The extraction of *R. felis* reads was performed using bwa-aln (-n 0.02 -l 1024) to later determine the aDNA authentication. The authentication of ancient DNA (aDNA) reads ascribed to these taxa was achieved by library-independent verification using mapDamage[29] and the analyses of the read-length distribution (bp). Subsequently, we mapped our dataset (bwa-mem) against all the currently available (i.e., 126) NCBI *Rickettsia* reference genomes, four of which comprise *R. felis* genomes. Following these steps, we were able to identify, at species level, 19,840 unique authenticated aDNA sequence reads mapping (bwa-aln) to the genome of *R. felis* strain LSU-Lb (SI 3), with the LPB, RPB and ULPM reads collectively providing 58.01-fold genome coverage against this reference genome.

**Ancient DNA authentication.** The authentication of aDNA sequence reads ascribed to *R. felis* was achieved by library-independent verification using mapDamage[29] (Fig. 2b) and analyses of both the read-length distribution (bp) (Fig. 2c) and edit-distances (Fig. 2d) ("Methods"). Consistent with the characteristics of aDNA, we detected significant DNA damage patterns for the reads mapping to the *R. felis* de-novo genome assembly (SI 4). The mean read-length distribution of all BBayA *R. felis* datasets (84.57 bp) furthermore indicated that the DNA was in a highly fragmented state. Damage pattern and read-length distribution analysis of the host (BBayA) DNA exhibited

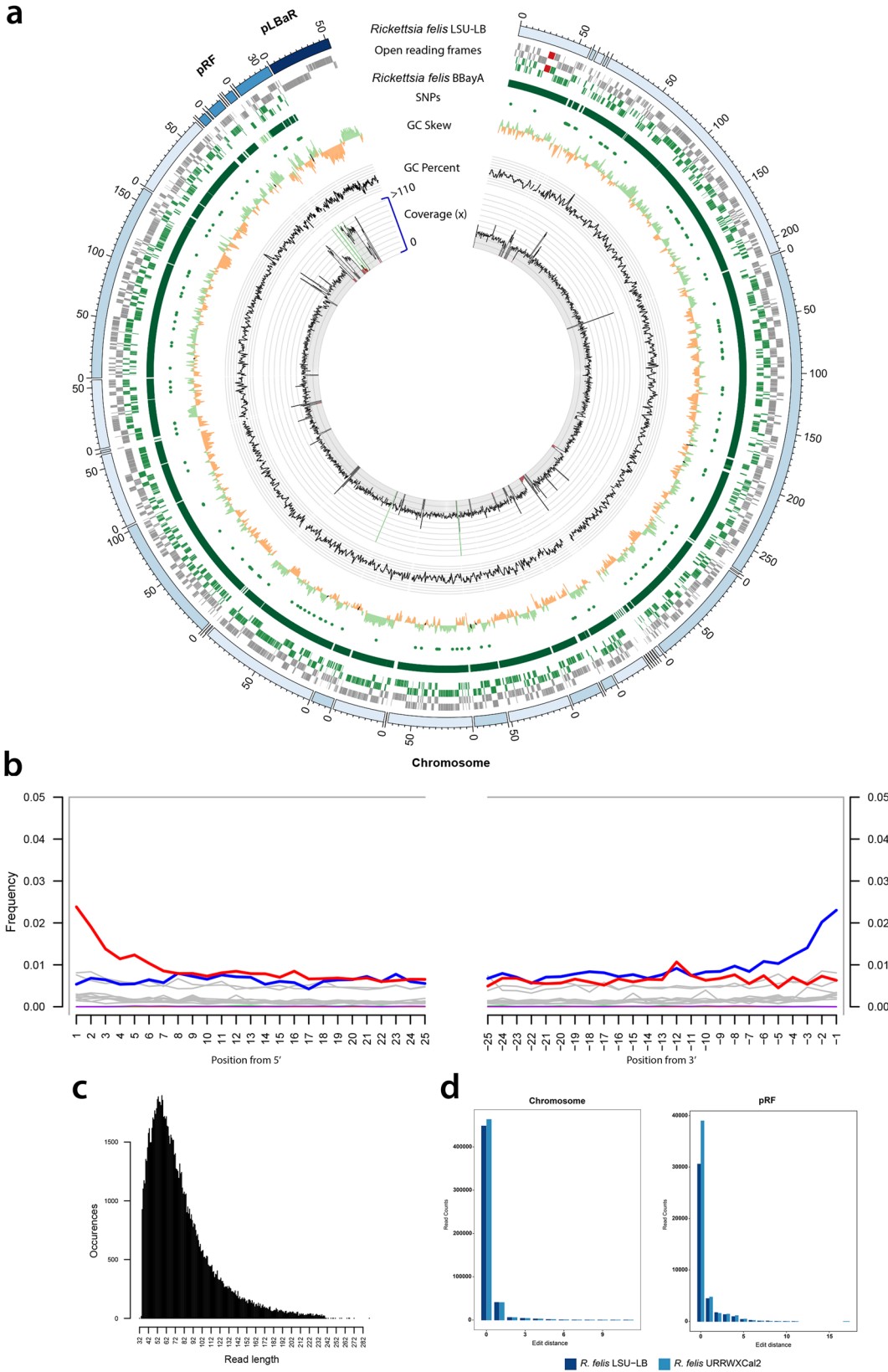

a similar DNA damage profile and short (i.e., damaged) sequence read-length distribution (Fig. S1).

**Ancient genome reconstruction**. To confirm that the organism represented in our metagenomic output was an *R. felis* strain, and not a closely-related southern African species (e.g., *R. prowazekii*, *R. typhi*, *R. conorii* and *R. africae*), and to detect signs of plasmid rearrangements, we mapped our datasets (bwa-mem) against all currently available (i.e., 126) NCBI *Rickettsia* genomes (Supplementary Data 3). *Rickettsia* genomes comprise 1.1–1.8 million base pairs (Mbp) and exhibit a high percentage of non-coding

**Fig. 2 Genome reconstruction of the ancient BBayA R. felis genome. a** Genome reconstruction of the ancient BBayA *R. felis* genome and mapping of the ancient genome to the genome of *R. felis* LSU-Lb. The *R. felis* BBayA genome consists of 1,512,774 bases and 69 contigs, with a N50 of 42,410 bases and the longest contig comprising 121,989 bases. It exhibits a GC value of 32.5% and a coding density of 84.58% for 15161 predicted proteins. A comparison was performed with the chromosome and plasmids (*pRF* and *pLBaR*). Rings (from outer to inner ring) show ORFs, SNPs, GC skew, GC content and coverage. Following the initial mapping of our datasets against all available (i.e., 126) NCBI *Rickettsia* reference genomes (Supplementary Data 3), the genome coverage analysis of BBayA *R. felis* was performed using the reads mapping to the *R. felis* LSU-Lb genome (the closest phylogenetic homologue to the ancient BBayA *R. felis* strain), with average coverage estimated at 58.01-fold. We estimated that 94.73% of the LSU-Lb genome was covered by the BBayA *R. felis* reads. The RHS-like toxin plasmid (*pLBaR*-38) occurs only in *R. felis* LSU-Lb, with *R. felis* URRWxCal2 and the ancient BBayA being devoid of it. The coverage for the LSU-Lb *pRF* plasmid within the BBayA read dataset is 132.26 (mean) and 106.19 (trimmed mean), and 0/0 for *pLBaR*. **b** DNA damage pattern analysis for the BBayA *R. felis* reads using mapDamage. G-to-A and C-to-T misincorporations are plotted in blue and red, respectively, and grey lines indicate all possible misincorporations. **c** DNA fragment read-length distributions of the BBayA *R. felis* reads, exhibiting a mean read-length of 82.30 base-pairs (bp). **d** Distribution of edit distance of high quality BBayA *R. felis* reads mapping to *R. felis* LSU-Lb and *R. felis* URRWxCal2.

DNA, indicative of a process of reductive evolution[30]. The plasmid system in *R. felis* is unusual, since no other bacteria in the Rickettsiales (i.e., *Anaplasma*, *Neorickettsia,* and *Wolbachia*) are known to harbour plasmids. We were able to recover reads homologous to 94.73% of the *R. felis* LSU-Lb genome (bwa-mem-min-read-percent-identity 95-min-read-aligned-percent 50), while the ancient *R. felis* BBayA genome assembly is 99.53% complete according to CheckM software (Materials and Methods). We calculated the coverage of the BBayA *R. felis* genome over mapped reads used for the assembly being 55.1-fold, representing a trimmed value following removal of 5% from both extremes (i.e., low and high coverage) across the genome. The mean coverage without the trimming values was 61.7-times, including in both cases the pRF plasmid (Fig. 2a). Phylogenetic analysis revealed that the *R. felis* LSU-Lb strain is the closest homologue to the ancient BBayA *R. felis* strain described here.

**Phylogenetic placement of the BBayA *R. felis* genome**. The assembly of the BBayA *R. felis* genome resulted in the recognition of the single *Rickettsia* chromosome and the detection of one plasmid, i.e., *pRF* (Fig. 2a and Supplementary Data 4). Phylogenetic analyses of the BBayA *R. felis* genome revealed strong clustering within the recently proposed *R. felis* transitional group *Rickettsia* (TRG), which is characterised by including both vertebrate *Rickettsia* and *Rickettsia* infecting non-blood feeding arthropods (Fig. 3a). We further note that BBayA *R. felis* genome is closely affiliated with three other *R. felis* genomes including the *R. felis* URRWxCal2 reference genome (Fig. 3c), showing consistency within the species. BBayA *R. felis* also exhibits close affinities to the better-known and highly pathogenic *R. typhi* (the causative agent of murine typhus) and *R. prowazekii* (the aetiologic agent of epidemic typhus) from the typhus group (TG) rickettsiae (Fig. 3b). Given a lack or temporal signals in the ancient DNA dataset, we were unable to determine chronometric stages in the evolution of the BBayA *R. felis*, including the emergence of a most recent common ancestor (MRCA) for the southern African *R. felis* group (Fig. S2) (SI 5) ("Methods").

When compared to the *R. felis* (i.e., LSU-Lb and URRWxCal2) and other *Rickettsia* genomes used in this study (i.e., *R. typhi*, *R. prowazekii* and *R. africae*), several SNPs are specific to the BBayA *R. felis* strain (Supplementary Data 4). One missense variant (mutation) was identified in the *cell surface protein 2* (Sca2) coding region of *R. felis* LSU-Lb, but was absent in URRxCal2. Sca2 (*pRF*25) was detected on the BBayA *R. felis* pRF plasmid. It is a noteworthy virulence protein in *Rickettsia* as it facilitates cell adherence[31] and promotes pathogenesis in primary and secondary hosts. The plasmid *pLBaR*, which is absent in the BBayA *R. felis* genome but present in the *R. felis* LSU-Lb genome, encodes a repeats-in-toxin-like type I secretion system and an associated RHS-like toxin, namely *pLbaR*-38. No other deletions or insertions were detected in the BBayA *R. felis* genome.

**Discussion**
The implications of the molecular detection of a 2000-year-old bacterial pathogen, in association with a hunter-gatherer child from southern Africa, is significant. Formerly, the identification of skeletal pathologies presented the only means by which information concerning the incidence of diseases in the archaeological record could be gained. However, morphological analyses are limited as not all pathogenic infections necessarily result in diagnostic skeletal lesions[32]. It has since been verified that the DNA of pathogenic bacteria, such as *Brucella melitensis*[32], *Mycobacterium leprae*[33], *M. tuberculosis*[34], *Yersinia pestis*[35], certain ancient *Salmonella enterica* serovars[36] and *Borrelia recurrentis*[37], viruses, such as Hepatitis B virus[38], and parasitic organisms including *Plasmodium falciparum*[39], can be retrieved from ancient human skeletal remains. Here, we add *R. felis* to the list of pathogens than can be recovered from ancient human remains. We furthermore demonstrate that the DNA of ancient pathogenic microbial taxa can also be recovered from prehistoric sub-Saharan African human skeletal remains, and from human petrous bone samples (SI 6).

Since the DNA analysed in this study was extracted from bone (petrous, i.e., LPB and RPB) and tooth (upper premolar, i.e., ULPM) samples, the fact that not all pathogenic microbial taxonomic categories might be recoverable from either human skeletal or dental remains[39] suggest that some taxa might be underrepresented. In this regard, our study confirms that the DNA of *R. felis* can be recovered from human petrous bone. A previous stuy[39] focussed on the differential detection of a single pathogen, *Yersinia pestis*, from teeth and petrous samples, showing a higher microbial diversity in teeth than petrous bones, including additional pathogenic and oral taxa. The reasons cited for this result include the fact that the otic capsules of the petrous bones are harder than tooth roots, implying that very little exogenous DNA will penetrate into these bones. Differences in blood circulation and bone turnover rates in petrous regions and teeth may therefore account for the variable incidence of pathogenic DNA in these respective sample sites. *R. felis* is blood-borne and has been detected in the blood and cerebrospinal fluid of individuals diagnosed with malaria, cryptococcal meningitis and also scrub typhus (SI 7). In this instance, the remains analysed represented that of a child, the skull and teeth of which were still experiencing formative development and, therefore, not yet fully fused, developed and densified. In addition, chronic diseases and resulting comorbidities are associated with diminished bone mineral accrual and bone loss, and various paediatric disorders have been implicated in impaired bone health[40]. It is therefore probable that BBayA displayed irregular and abnormally-low skeletal bone density and skeletal metabolism, in turn resulting in an increasing predisposition of pathogenic microbes to circulate through and enter the dense otic capsules of the petrous regions. Consequently, this resulted in the extraction of an assembled genome with >99% completeness (according to the CheckM

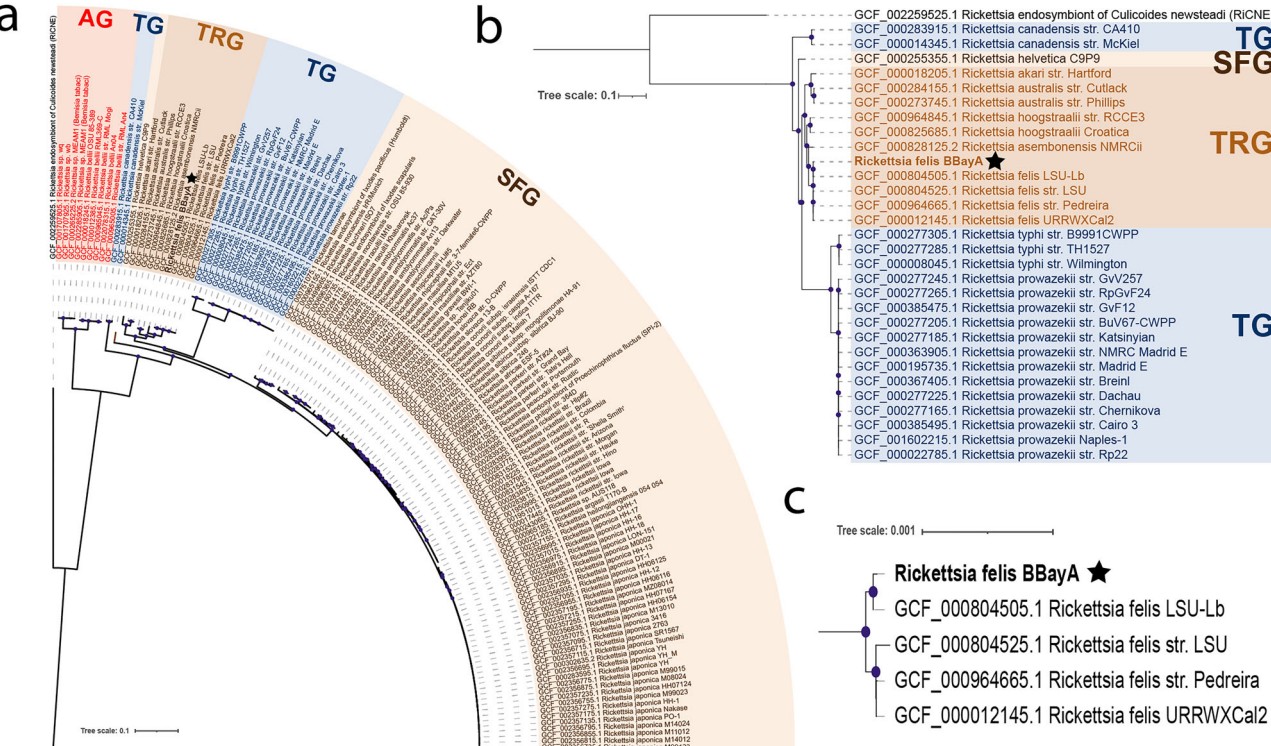

**Fig. 3 Phylogenomic reconstruction of the Rickettsia genus including the ancient R. felis strain derived from BBayA. a** Maximum likelihood (ML) phylogenomic tree including 127 *Rickettsia* genomes, **b** pruned ML phylogenomic tree including 31 genomes from the TG and TRG *Rickettsia* groups and the five *R. felis* genomes and **c** inset of the *R. felis* clade. Purple dots in leaf nodes represent confident ultrafast bootstrap support values of >95%. Multiple sequence alignment of the codons of 138 concatenated core genes for the 127 *Rickettsia* genomes was used for the phylogenomic reconstruction using the maximum likelihood method (ML). The ML reconstruction was obtained with the IQtree software using the best substitution model (GTR + F + R10) selected using the ModelFinder option and utilising all the sites from the codon alignment of the 138 selected core genes (103,280 nucleotide sites), from which 53.4% of the sites were constant across the 127 genomes (55,120 nucleotide sites). The parsimony informative sites were 33,343 nucleotide sites which present 17,683 distinct nucleotide composition patterns. The legend displays the branch length. The black star (★) indicates the position of the *Rickettsia felis* BBayA genome. The trees indicate the four *Rickettsia* groups. i.e., the spotted fever group (SFG), typhus group (TG), transitional group (TRG) and ancestral group (AG) (see Fig. S4 and Supplementary Data 6 for additional maximum-likelihood trees constructed using the same codon alignment). The final resulting tree was analysed in the iTol tree visualisation tool[82], with phylogenetic reconstructions visualised and managed using the iTOL web server[83].

software) of ancient *R. felis* from reads analysed in this study, from the RPB and LPB samples. In addition to the young age and compromised health of the child, taphonomic factors might further explain the differential preservation of microbial DNA in the petrous and tooth (ULPM) samples. The context from which the child's remains were retrieved comprised shell midden overlooking the beach, ~ 46 m from the high-Indian Ocean water mark. The humid saline conditions and loose sedimentary matrix may certainly have resulted in increasingly rapid DNA degradation, particularly in the exposed sub-adult teeth[41]. Ultimately, the sequencing strategy originally employed resulted in the sequencing of seven libraries derived from the left (LPB) and right (RPB) petrous samples, and only a single library from the upper-left premolar (ULPM), introducing a bias in terms of the numbers of microbial reads recovered from the respective samples.

Osteobiographic analysis[25] is consistent with the premise that various chronic and acute viral, bacterial, and parasitic infections could have produced the skeletal signs likely representing anaemia as observed in the child[42,43]. Indications of *cribra orbitalia* are a symptom of marrow expansion caused by haemopoietic factors, and has been attributed to both malnutrition (e.g., megaloblastic anaemia) and parasitism. However, other plausible causes for this pathology include malaria (*Plasmodium sp.*), hookworm infection (*Ancylostoma duodenale* and *Necator americanus*) and schistosomiasis (*Schistosoma haematobium*), the latter of which was suggested as the best-fit cause for the child's pathology. Besides

*R. felis*, which cause comparable osteological pathologies[44], the pathogens referred to above were absent from our dataset. *R. felis* is an obligate intracellular pathogen which, in order to establish productive infections, also modifies the cytoskeletal architecture and the endomembrane system of their host cells[44,45] (SI 7).

We cannot detect, with certainty, changes in the virulence or host specificity, over the past ~2000 years of evolutionary history, of *R. felis* in southern Africa. The observed variation that exists between extant *R. felis* genomes may represent transient genetic fluctuation, while the evolutionary relevance of which is still uncertain[41]. The small genomes of *Rickettsia* (1.1–1.8 million bp), and a high percentage of non-coding DNA[30] may also explain the limited divergence observed. With regards to the pathogenicity of the BBayA *R. felis* strain, it is uncertain whether the absence of the RHS-like toxin plasmid (*pLbaR*-38) resulted in the ancient strain being less pathogenic to humans. Modern *R. felis* strains, in particular *R. felis* LSU-Lb which harbours the RHS-like toxin plasmid (*pLbaR*-38), are known to be highly pathogenic to human hosts. Similarly, it remains uncertain whether *R. felis* might have evolved to become more pathogenic, as is the case with certain *Salmonella enterica* serovars, e.g., *S. enterica* ssp. *enterica*, and which is associated with the cultural and economic transformations following the beginning of the Neolithic[46]. The presence of the Sca2 (*pRF*25) mutation suggests that this ancient BBayA *R. felis* strain was indeed pathogenic, a conclusion supported by the fact that, besides the pLBaR plasmid, the minimal

genomic divergence distinguishing *R. felis* LSU-Lb from flea-associated strains suggests that it has the potential to be a human pathogen[30]. The BBayA *R. felis* strain may therefore have resulted in symptoms typical of typhus-like flea-borne rickettsioses, including fever, fatigue, headache, maculopapular rash, sub-acute meningitis and pneumonia (SI 7).

*R. felis*, an insect-borne pathogen and the causative agent of typhus-like flea-borne 'spotted fever', is an obligate intracellular bacterium in the order Rickettsiales[47]. While cat- and dog-fleas (*Ctenocephalides felis* and *C. canis*) have been cited as the most probable vectors, ˃40 different haematophagous species of fleas, mosquitoes, ticks and mites have been identified as vectors[48]. As well as the identification of the African great apes (chimpanzees, gorillas, and bonobos) as vertebrate reservoirs responsible for the maintenance of *R. felis* in Africa, it has been proposed that humans are natural *R. felis* reservoirs[49], just as they are for certain *Plasmodium* species[50]. Its detection in 2000-year-old old human remains strongly supports this view. The clinical presentation of rickettsial diseases ranges from mild to severe. Without antibiotic treatment, murine or 'endemic' typhus, caused by *R. typhi*, exhibits a mortality rate of 4%, and Rocky Mountain spotted fever a mortality rate as high as 30%[51]. Epidemic typhus, caused by *R. prowazekii*, has a mortality rate which varies from 0.7 to 60% for untreated cases. Mortality rates as high as 66% has been reported for disease due to *R. rickettsii* occurring prior to 1920, preceding the discovery of antibiotics[50]. Human disease case fatality rates, the proportion of patients that reportedly died as a result of infection, of 19% have been reported for untreated *R. felis* infections[52]. In Africa, *R. felis* is the causative organism of many (~15%) cases of illnesses classified as 'fevers of unknown origin', including febrile seizures or convulsions[44]. Relative to TG (i.e., transmitted by body lice and fleas) and SFG (transmitted by ticks) rickettsiae, a much wider host range has been reported for TRG rickettsiae, including ticks, mites, fleas, booklice and various other haematophagous insects[31], including mosquitos of the genera *Aedes* and *Anopheles*[53,54]. In addition, similar to *R. typhi*, *R. felis* is also shed in flea faeces, providing an additional avenue for zoonotic host to human infection.

The emergence, in South Africa, of this particular *R. felis* strain may well relate to socio-demographic factors. Specifically, an increase in population density, driven by cultural change and technological innovation, may have resulted in more frequent instances of human *R. felis* infections during the Later Stone Age in southern Africa. As is the case amongst ethnographically-known Kalahari hunter-gatherers[55], ancient human social networks likely functioned to facilitate the aggregation of isolated hunter-gatherer bands and the maintenance of social relations, which generally transpired during environmentally-stressful conditions. Although geographically-dispersed, hunter-gatherer social networks have been shown to facilitate both the transmission and the persistence of various infectious, zoonotic and parasitic diseases. This would therefore have prevented a reduction in infection risk, which is generally expected to have occurred, amongst itinerant hunter-gatherer groups[56].

## Conclusion

Whereas the first account of a typhus-like disease appears in AD 1489, during the War of Granada[4], our findings provide novel baseline data concerning the incidence of a pathogenic microbe amongst ancient, pre-Neolithic, South African hunter-gatherers. *Rickettsia felis* can no longer be considered a novel or emerging pathogen that originated in the global north. Our results necessitate further discussion about the susceptibility of humans to, and the population impacts of, zoonotic diseases on human longevity and behaviour in the past. It is evident that, given the temporal depth of human occupation in sub-Saharan Africa, and

the preservation of aDNA in local archaeological contexts, the region is well positioned to play a key role in the exploration of ancient pathogenic drivers of human evolution and mortality.

## Methods

**aDNA sources and extraction.** The skeletal remains of Ballito Bay A ('BBayA') belong to a juvenile individual excavated during the 1960s. The remains were curated at the Durban Museum, and then transferred to the KwaZulu-Natal Museum where these are now curated (accession No. 2009/007)[2]. Permission for the sampling of the remains was obtained from the Council of the KwaZulu-Natal Museum. A sampling permit (No. 0014/06) was issued to M. Lombard under the KwaZulu-Natal Heritage Act No. 4 of 2008 and Section 38 (1) of the National Heritage Resources Act No. 25 of 1999. From the accessioned skeletal remains, analysed samples were extracted from the left petrous bone, right petrous bone and the upper left premolar (Table 1). Under the latter legislation, permits were issued by the South African Heritage Resources Agency (SAHRA) for the destructive sampling and ancient DNA analyses at Uppsala University, Sweden (No. 1939), and for sending samples for radiocarbon dating to Beta Analytic, England (No. 1940)[2]. The originally-published manuscript from which the genomic data analysed in this study derives, i.e., Schlebusch et al. 2017, is available at 'https://www.researchgate.net/publication/320101464_Southern_African_ancient_genomes_estimate_modern_human_divergence_to_350000-260000_years_ago'.

Prior to sampling, the bone samples were UV irradiated (254 nm) for 30 min to one hour per side and stored in plastic zip-lock bags until sampled[2]. Further handling of the specimens was done in a bleach-decontaminated, also using DNA-Away (Thermo Scientific) enclosed sampling tent with adherent gloves (Captair Pyramide portable isolation enclosure, Erlab). Teeth were wiped with 0.5% bleach (NaOH) and UV-irradiated sterile water (HPLC grade, Sigma-Aldrich). The outer surface was removed by drilling at low speed using a portable Dremel 8100, and between 60 and 200 mg of bone powder was sampled for DNA analyses from the interior of the bones and teeth. The researchers wore full-zip suits with caps, facemasks with visors and double latex gloves and the tent was frequently cleaned with DNA-Away during sampling. The 1.5 ml tubes containing the bone powder samples were thoroughly wiped with DNA-Away before they were taken into the dedicated aDNA clean room facility at Uppsala University[2]. The laboratory is equipped with an air-lock between the lab and corridor, positive air pressure, UV lamps in the ceiling (254 nm) and HEPA-filtered laminar flow hoods. The laboratory is frequently cleaned with bleach (NaOH) and UV-irradiation and all equipment and non-biological reagents are regularly decontaminated with bleach and/or DNA-Away (Thermo Scientific) and UV irradiation. DNA was extracted from between 60 and 190 mg of bone powder using silica-based protocols[57] with modifications[58,59], and were eluted in 50–110 µl Elution Buffer (Qiagen). Between 3 and 6 DNA extracts were made for each individual (or accession number) and one negative extraction control was processed for every 4 to 7 samples extracted. The optimal number of PCR cycles to use for each library was determined using quantitative PCR (qPCR) in order to see at what cycle a library reached the plateau (where it is saturated) and then deducting three cycles from that value. The 25 µl qPCR reactions were set up in duplicates and contained 1 µl of DNA library, 1X Maxima SYBR Green Mastermix and 200 nM of each IS7 and IS8 primers and were amplified according to supplier instructions (ThermoFisher Scientific)[60]. Each library was then amplified in four or eight reactions using between 12 and 21 PCR cycles. One negative PCR control was set up for every four libraries. Blunt-end reactions were prepared and amplified using IS4 and index primers[57,60]. Damage-repair reactions had a final volume of 25 µl and contained 4 µl DNA library and the following in final concentrations; 1X AccuPrime Pfx Reaction Mix, 1.25U AccuPrime DNA Polymerase (ThermoFisher Scientific) and 400 nM of each the IS4 and index primers[60]. Thermal cycling conditions were as recommended by ThermoFisher with an annealing temperature of 60 °C[61]. The resulting libraries were quantified either on a TapeStation using a High Sensitivity kit (Agilent Technologies) or using a Bioanalyzer 2100 and a High Sensitivity DNA chip (Agilent Technologies). Regrettably, and given that the DNA extraction controls did not yield any DNA, and were therefore not sequenced[2], it is not possible to include any information regarding the analyses of taxa detected in negative controls in this study, although this is standard practice in aDNA-related research. The DNA libraries were sequenced at SciLife Sequencing Centre in Uppsala using either Illumina HiSeq 2500 with v2 paired-end 125 bp chemistry or HiSeq XTen with paired-end 150 bp chemistry. The initial strategy was to screen the DNA extracts to evaluate the endogenous ancient human DNA content by building blunt-end libraries and sequencing each library on either a 1/10th of a HiSeq 2500 lane or on a 1/20th of a HiSeq XTen lane. Additional blunt-end or damage-repair libraries were then built, and sequenced and high-quality libraries were sequenced to completion (up to 97% clonality) while libraries with low endogenous contents were sequenced to a lesser extent (average 36% clonality over all libraries)[2].

**Authentication of ancient pathogenic DNA.** Following the application of bioinformatic analytical protocols, the resultant data-set indicated the presence of a single authentic (ancient) pathogenic taxon subjected to and verified according to the

**Table 1 Authenticated ancient DNA sequence reads, derived from eight aDNA shotgun metagenome sequence libraries generated from the boy from Ballito Bay (BBayA), mapped (bwa-aln) to the *Rickettsia felis* LSU-Lb reference genome.**

| | Sample source | Libraries | Total reads | Human reads | *Rickettsia* reads | % of reads | % duplicates | Unique *R. felis* reads | Mean read length (bp) | Genome coverage (mean) |
|---|---|---|---|---|---|---|---|---|---|---|
| *Rickettsia felis* | Left petrous bone | 5 | 3768601170 | 491223634 | 525855 | 0.0140 | 37.53 | 13477 | 84.40 | 18.4353 |
| | Right petrous bone | 2 | 1075114194 | 244053412 | 250195 | 0.0233 | 25.57 | 6360 | 84.90 | 10.5154 |
| | Upper left premolar | 1 | 42434936 | 18050 | 562 | 0.0013 | 12.09 | 83 | 83.00 | 0.0001 |
| | Total | 8 | 4886150300 | 735295096 | 776612 | 0.0159 | 33.66 | 19840 | 84.57 | 28.9735 |

authentication process outlined[2]. Briefly, molecular damage accumulating after death is a standard feature of all aDNA molecules. The accumulation of deaminated cytosine (uracil) within the overhanging ends of aDNA templates typically results in increasing cytosine (C) to thymine (T) misincorporation rates toward read starts, with matching guanine (G) to adenine (A) misincorporation rates increasing toward read ends in double-stranded library preparations[62]. Being the 'gold-standard' of aDNA authentication, we used mapDamage v2.0.1[29] to determine the incidence of cytosine (C) to thymine (T) and guanine (G) to adenine (A) substitution rates at the 5′-ends and 3′-ends of strands[62]. Damage un-repaired sequence libraries were used for the mapping to the *Rickettsia felis* and *Homo sapiens* reference genomes using BWA aln -n 0.02 -l 1024 parameters. Next, exact duplicate reads were removed using the MarkDuplicates (Picard) and the resulting alignment was used for the DNA damage analysis using the MapDamage tool[29] (https://academic.oup.com/bioinformatics/article/27/15/2153/404129). Mapped reads from the repaired and non-repaired libraries against the LSU-Lb genome were also analysed for damage patterns using PyDamage v0.70 software[63]. Accordingly, 36.90% and 60.76% of the mapped BBayA genome and the *R. felis* LSU-Lb genome, respectively, was authenticated as aDNA according to the strict *q*-values (<0.05), with an accuracy >0.5 for the test. As a substantial portion of the assembled genome could be authenticated as composed of ancient DNA, we are confident the genome assembled is ancient and not a result of recent contamination.

**Sequence data processing and analysis.** Paired-end aDNA sequencing reads were first processed to facilitate the removal of adapters and primers using Adapter-Removal v2[64] following the parameters 'min-quality' 20, 'min-length' 35 and 'collapsed to merge' the forward- and reverse-sequence reads. Human (*i.e.*, *H. sapiens*) reads were removed using the BWA-MEM algorithm against the human reference genome[61]. Using the new option '-preserve5p' with AdapterRemoval 2.3.1 (https://github.com/MikkelSchubert/adapterremoval/issues/32#issuecomment-504758137) resulted in a comparable DNA damage plot (Fig. S3). Kraken2 analysis[28] was performed using a custom database (including selected bacterial, archaeal, protozoa, and viral taxa) derived from the NCBI RefSeq database (https://www.ncbi.nlm.nih.gov/refseq/) with a high confidence (*i.e.*, 'cut-off' level) value of 0.85 to obtain the most accurate taxonomic assignments. The identification of microbial taxa is based on the use of exact-match database queries of *k*-mers, instead of alignment similarity. As different '*k*' values approximate degrees of taxonomic similarity, with $k = 21$ indicative of genus-level similarity, $k = 31$ of species-level similarity and $k = 51$ of strain-level similarity, we applied the default *k* value setting of 35 (i.e., $k = 35$). Using these results, pathogenic taxa were identified, and their respective reference genomes downloaded from the NCBI RefSeq database for the downstream analysis. Competitive alignment with Bowtie2[65] (-very-sensitive mode) was performed using the eight BBayA aDNA sequencing libraries (i.e., the 'petrous left', 'petrous right' and 'premolar' DNA sample libraries). Exact duplicates were removed using MarkDuplicates (Picard) (https://gatk.broadinstitute.org/hc/en-us/articles/360037052812-MarkDuplicates-Picard-).

**Genome reconstruction and comparative analysis of BBayA *R. felis*.** The *R. felis* LSU-Lb and URRWXcal2 strains were used as reference genomes during the BWA v0.6.2-r126 alignment (bwa aln -n 0.02 -l 1024) to the BBayA *R. felis* chromosomes and plasmids. FASTQ reads were extracted from the alignment and de-novo assembly was performed using the SPAdes v3.11 genome assembler at default parameter settings (http://cab.spbu.ru/files/release3.11.1/manual.html#correctoropt)[66]. The assembled ancient *R. felis* genome was used for average nucleotide identity and single nucleotide variant analysis using FastANI[67] and kSNP3 v3.1[68]. This involved first identifying the optimum *k*-mer value for the all the selected *Rickettsia* genomes using K-chooser, with the value 31 identified the best *k*-mer value for the kSNP analysis. kSNP analysis was then performed using the 'kSNP3 -in fasta-input-list -ML -CPU 4 -outdir kSNP-out -k 31 -annotate annotate-list'. The pan-genome analysis was performed and the core genes identified using 126 *Rickettsia* genomes and the BBayA *R. felis* genome with the GET_HOMOLOGUES package[69] using default parameter settings (https://github.com/eead-csic-compbio/get_homologues). The genome comparison and coverage plots were visualised using Circos[70] (http://circos.ca/). The quality of all the *Rickettsia* genomes was evaluated with the CheckM v1.1.3 software package[71] (Supplementary Data 5). Of the available NCBI RefSeq genomes used, some displayed lower completeness values (e.g., 90.97% for GCF_000964995.1) and higher contamination values (e.g., 7.04% for GCF_000696365.1). The BBayA genome completeness was 99.53%, and contamination 0.47% according to CheckM software. From the other *R. felis* strain genomes, only those from the strains Pedreira and URRWXCal2 achieved 100% completeness, with 0% and 0.47% contamination, respectively. The remaining two *R. felis* strains, namely LSU-Lb and LSU, presented lower completeness and contamination values (at 97.47% and 1.42%, and 94.14% and 0.71%, respectively (Supplementary Data 3)). While our assembly is dependent on the mapped reads against reference genomes, unknown regions of the ancient genome could have been lost, and further studies are needed in this regard. Following this, the reads that were used to assemble the *R. felis* genome were mapped against the obtained assembly, as well the other 126 *Rickettsia* genomes, using Coverm v0.6.1 (https://github.com/wwood/CoverM) with the BWA-MEM mapping tool, resulting in a) a competitive mapping with the five *R. felis* genomes recruiting over 80% of reads, b) in a non-competitive mapping (bwa-mem -min-

read-percent-identity 95 -min-read-aligned-percent 80), the percentage of reads used in the BBayA assembly were at least >89% for any of the *R. felis* genomes, while numbers <72% were found for the other species, and c) the percentage of bases of the genome covered for at least 1 read at 95% identity and 80% of read alignment, was >98% for the *R. felis* genomes, excepting the *R. felis* str. Pedreira with 94%, GCF_000964665.1, while for other species it was <80% (e.g., GCF_000828125.2, *R. asembonensis*). As indicated, our ancient *R. felis* BBayA genome consists of 1,512,774 bases and 69 contigs, with a N50 of 42,410 bases and the longest contig comprising 121,989 bases. It exhibits a GC value of 32.5% and a coding density of 84.58% for 15161 predicted proteins.

**Phylogenetic analysis of BBayA *R. felis*.** A concatenated codon alignment was produced from 138 protein sequences (the core genes identified by the GET_HO-MOLOGUES package as previously described) from all the 127 available *Rickettsia* strains. Each protein alignment was performed using MAFFT v7.46476 at default parameter settings (https://mafft.cbrc.jp/alignment/software/), therefore alignments were reverse-transcribed to the codons using PAL2NAL v14 software[72], alignment blocks were obtained using Gblocks 0.91b software with default parameters[72] and were concatenated using custom scripts. Maximum-likelihood tree reconstruction was performed with IQtree v.1.5.5 software[73–75] for the obtained codon alignment (103,280 nucleotide sites) with the TESTNEW option to select the best substitution model (GTR + F + R10 according to ModelFinder)[76] and with a non-parametric ultrafast bootstrap (-bb) test of 10,000 replicates. Phylogenetic reconstructions were visualised and managed using the iTOL web server[33]. Maximum-likelihood trees were also constructed using the same codon alignment with the FastTree version 2.1.10[77] with -gtr and -gamma options, the RAxML version 8.2.11[78] with -m GTRGAMMA, -#100 (to search the best tree between 100 replicates) and -# autoMR option to determine node bootstraps with automatic number of replicates; and MEGA-CC version 11.0.10[79] using GTR (G + I) model and bootstrap support of 100 replicates (see Supplementary Data 6). These phylogenomic reconstructions were compared using the approximately unbiased (AU) test[80] implemented in IQ-TREE v.1.5.5 with the options -n 0 -zb 10000 -au -zw. The *p*-values for the AU test of the FastTree (*p*-value 0.337), IQtree (*p*-value 0.727) and RAxML (*p*-value 0.315) reconstructions indicated these trees as 95% confident sets, while the MEGA-CC tree got a significant exclusion (*p*-value 0.000127) (Fig. S4). Regarding the phylogenetic placement of BBayA *R. felis* and its closeness to the TG group, we performed a clustering analysis based on MASH and FastANI values using dRep v3.2.2[81]. This confirmed that the TRG clade, into which *R. felis* is classified, is closer to the SFG clade than the TG clade. Notably, the MASH average nucleotide identity values of TRG were ~92% with SFG and ~87% with the TG group.

**Reporting summary**. Further information on research design is available in the Nature Portfolio Reporting Summary linked to this article.

## Data availability

Raw reads from Ballito Bay A samples are available under the NCBI BioProject PRJEB22660. The *R. felis* BBayA mapped reads and the metagenome-assembled genome are available under the NCBI BioProject PRJNA930765. The NCBI WGS accession number is JAQQRK000000000.

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

## Acknowledgements

RFR acknowledges the funding provided by a National Geographic Society Scientific Exploration Grant (No. NGS-371R-18) and by the Oppenheimer Endowed Fellowship in Molecular Archaeology (the Benjamin R. Oppenheimer Trust). CMS is funded by the European Research Council (ERC) under the European Union's Horizon 2020 Research and Innovation Programme (Grant Agreement No. 759933) and the Knut and Alice Wallenberg Foundation. We thank Yves Van de Peer, Stephane Rombauts (Bioinformatics and Evolutionary Genomics Group, VIB-UGent, Ghent, Belgium) and Ansie Yssel (BGM, CMEG, University of Pretoria, Pretoria, South Africa) for analytical support. Sequencing was performed at the SNP&SEQ Technology Platform, SciLife Lab, National Genomics Infrastructure, Uppsala, and computational analyses were performed at the Centre for Microbial Ecology and Genomics (CMEG), University of Pretoria, South Africa.

## Author contributions

R.F.R., M.L., and J.B.R. conceived the study and composed the manuscript. S.V., J.A., and J.B.R. performed the bioinformatic and statistical analyses, and R.F.R., M.L., J.A., and S.V. generated the figures. C.S., M.J., and M.L. generated the sequence datasets and D.A.C. provided access to analytical facilities. The KwaZulu-Natal museum provided access to the human remains in terms of sampling, export and dating permits issued to ML (#s 0014/06, 1939, 1940) according to the KwaZulu-Natal Heritage Act No. 4 of 2008 and Section 38 (1) of the National Heritage Resources Act No. 25 of 1999. Final reports have been submitted to the repository and both heritage agencies. All authors contributed to the completion of the final manuscript.

## Competing interests

The authors declare no competing interests. The funding sponsors had no role in the design of the study, the collection, analyses, and interpretation of data, in the writing of the manuscript or in the decision to distribute the results.
