## [Peer Review File · Communications Biology]

Reviewers' comments:

Reviewer #1 (Remarks to the Author):

Review for Ancient DNA of *Rickettsia felis* and *Toxoplasma gondii* implicated in the death of a hunter-gatherer boy from South Africa, 2,000 years ago

The authors describe the detection of ancient DNA reads and genome reconstruction of *Rickettsia felis* from a 2,000 year old boy from South Africa. They also identify reads belonging to *Toxoplasma gondii*, as well as *Anaplasma*, *Babesia*, *Bordetella*, *Brucella*, *Leishmania*, *Plasmodium*, and *Trypanosoma*. The authors use that information to make inferences on the causative agent of the boys' death, and the evolutionary timeline of *R. felis* and *T. gondii*.

Major points

In general, there is a large amount of unclear language in the paper. A few examples are: lines 32 - 34. For someone unfamiliar with the cited papers, this sentence is unclear. How was it determined that the boy was anaemic? And what is meant by 'revised the genetic time-depth for *Homo sapiens*'? I think it'd be good if the unfamiliar reader wouldn't have to look at two references, given that this is the first sentence of the paper. Also, figure 1 is unrelated to the sentence. line 21 - 22 and also line 78 - 79: '... there is insufficient evidence for ancient human-pathogen interaction in the region.' This is unclear. 'Interaction' can mean all sorts of things. 'infection' may be a better choice of word. lines 34 - 36. The language in that sentence is unclear. The authors write that 'associated genetic data are generated', without specifying to what that data is associated too. One can assume that the authors mean the human genome, but this should be explicit. line 36, next sentence. 'These data contain traces of...' The authors may wish to be more cautious here, since not all samples always contain traces of potentially pathogenic microbes. the title of SI9 doesn't reflect the content of the text in SI9. There are also a number of points that are made in the introduction that would be better to make in the discussion, such as line 42: 'A co-infection of these...' and line 47: 'Our results show that these, and various other...'. A number of references to the biology of the detected pathogens (line 123ff, 127ff, 268ff) are made throughout the manuscript. I think it'd be better if they could be grouped together. References to authentication of reads (lines 81ff, 91ff, 152ff, 216ff) could likewise be grouped in one place. There is also a lot of text provided that is not really necessary for the argument of the paper, such as the whole paragraph from line 51ff, and 62ff. I realise that some of these issues are a matter of taste and do not impact the findings of the paper, so I leave it to the editor to decide if these should be fixed.

The authors described that after taxonomic classification with Kraken2, they competitively mapped against a selected set of reference sequences of pathogenic taxa as well as the complete human genome. However, this carries the risk that reads that map best against *R. felis*, or another of their chosen references, would actually map better against some other sequence that's not in the selected set of references. The authors should map the reads that map against any of the selected references against the whole nt database, to ensure that the reads do not in fact map better to a different reference, otherwise there's little confidence that the reads are assigned to the correct species or genus. I also have concerns that since the reads mapping against *T. gondii* map mainly in repeat regions and non-coding regions (line 220ff), those may in fact be off-target matches.

The authors identified reads mapping against *R. felis* and *T. gondii* at the species level, as well as *Anaplasma*, *Babesia*, *Bordetella*, *Brucella*, *Leishmania*, *Plasmodium*, and *Trypanosoma* at the genus level. However, the text doesn't make it clear why they chose to focus on *R. felis* and *T. gondii*, even though both *Bordetella* and *Brucella* have higher coverage depths, that could possibly allow for the assembly of a (partial) genome. Finally, I'm not sure about choice of the *R. felis* reference genome. The authors mapped against all available *R. felis* genomes (line 118ff), to exclude that the reads don't represent another related southern African *Rickettsia* species. What information was gained by

mapping against all those *R. felis* genomes? The authors should map against all other related southern African *Rickettsia* as well, to ensure that the highest number of reads, with the highest score is actually against *R. felis*. On line 127 the authors say 'Their unique genomic structure nevertheless allows high mapping specificity...' I don't follow. What is unique about the genomic structure, and why does that allow for high mapping specificity? The authors further state that the level of coverage allowed them to infer the presence and absence of genomic regions, but they don't say what level of coverage they required to do so. The authors determine that BBayA *R. felis* is closest to *R. felis* LSU-Lb on the basis of phylogenetic analysis (line 134). However, that statement is premature in the text, and from the way the phylogenetic analysis is presented in the manuscript, the higher similarity to *R. felis* LSU-Lb is not apparent. Also, looking at figure 2d, the edit distances are smaller against *R. felis* URRWxCal2 than LSU-Lb for the chromosome and pRF. The possibility that BBayA is actually closer to *R. felis* URRWxCal2 and has lost pLBaR over the last ~2,000 years is not discussed.

The phylogenetic analyses need to be re-considered. The topology of the tree in figure 3 and figure S3, in particular the placement of the *R. helvetica* C9P9 strain is different. The colouring of the TG and TRG groups is different in the two figures (*R. australis*, *R. akari*). Posterior probabilities and support values should be shown for Fig S3 and 3, respectively. Figure S3 should show the actual positions of the BBayA, URRWxCal2 and LSU-Lb sequences, since the authors refer to those divergence dates in the text (line 180ff, line 272ff). On line 272ff, the authors also mention that the BBayA *R. felis* strain pre-dates 5000 years, which is not clear from the figure. The authors should indicate why they selected the 138 genes that were used for the analysis. The labels on the axis in figure S3 should show less digits after the comma. The authors interpret the tree in figure 3 by saying 'the ancient BBayA *R. felis* genome occurs between *R. felis* LSU and *R. felis* LSU-Lb within the transitional group (TRG) rickettsiae, with close phylogenetic affinities to the reference *R. felis* URRWxCal2 reference genome.' 'between' is not a term generally used to describe phylogenetic positions. The tree in figure 3 shows a monophyletic clade consisting of BBayA *R. felis*, *R. felis* LSU, *R. felis* LSU-Lb, *R. felis* URRWxCal2, and *R. felis* Pedreira, with no difference between the five strains, in which case the vertical position of the strains has no meaning. If information is lost due to short branch lengths, the authors may wish to consider showing that part of the tree as an inset. Most importantly, molecular dating analyses can only be performed if enough temporal signal is present in the data (see <https://doi.org/10.1093/molbev/msv056> and <https://doi.org/10.1016/j.tree.2015.03.009>), and since only one ancient sequence is available, I'm unsure if this is the case. The authors should perform date randomisation tests, and regressions of root to tip distance and sampling dates to ensure that any temporal signal is present before performing molecular dating analyses.

The authors place a large emphasis on detecting the causative agent(s) of the boy's death (title, line 30, 42-43, 225ff, 255ff). They also suggest that the co-infection in particular exacerbated the severity of the boy's condition (line 225ff, 233ff). However, given that 2000 years have passed, and nothing is known about the circumstances of the death, and little is known about the effect of co-infections even in modern times, I find the wording in the main text very strong. Furthermore, the text in the main paper is in direct contradiction to the text in the well-argued SI 9 section, which provides seven reasons as to why it is difficult to determine the impact of the pathogens that they putatively identified on the boy's death. I suggest that the authors include some of the information in SI9 in the main text.

Minor points

line 39. It would be helpful to state here how complete the genome was.

line 83: What is meant by 'partially mapping reads'? Why were reads mapping to *T. gondii* mapping partially, and reads mapping to *R. felis* not? How was this determined?

Anaplasma is part of the Rickettsiales. Given this, what do the reads mean that are assigned to the *Anaplasma* at genus level? Should they also be assigned to *Rickettsia*? Or do the authors think there's a second strain of *Rickettsia* present apart from *R. felis*?

line 155. I don't see how edit distances against two *R. felis* strains help with showing that the reads are ancient.

What is meant by 'Because *R. felis* displays genotypic and phenotypic attributes of both spotted fever (SFG) and group (TG) rickettsiae, it is difficult to place phylogenetically.' (line 160ff) How was this determined? The trees don't show support values or posterior probabilities, so the reader can't judge. line 168: shouldn't this refer to Figure 3/S3? Also, the sentence specifically mentions BBayA having a close relationship to *R. prowazekii* and *R. typhi*, which is somewhat misleading, since the same thing can be said about the entire *R. felis* clade.

line 225: SI7 doesn't mention co-infection.

line 272: 'Here, we have demonstrated that the BBayA *R. felis* MRCA pre-dates 5,000 ya' This is unclear. Earlier, the authors state that *R. felis* LSI-Lb and BBayA *R. felis* have an MRCA of 2,000 ya and BBayA *R. felis* and *R. felis* URRWxCal2 have an MRCA of 3,000 years (line 180ff). It is therefore unclear where the 5,000 number comes from. If this refers to the MRCA of the entire *R. felis* clade, this has to be mentioned instead.

Line 273: Further, the author say that 'the strain .. diverged locally from the other TRG *R. felis* strains ~3,000 years ago'. How do they know this happened locally, and what is meant by that (locally as in local to Africa / South Africa etc)?

line 275ff / 286ff: '... no major changes in either the virulence or host specificity...' I don't think a single genome gives much information about host specificity and virulence. First, while 'multiple' SNPs specific to BBayA are detected, the authors only discuss one of them (line 207ff). Based on how difficult it is to link genotype to phenotype, I would refrain from making such strong statements. I also think talking about host specificity is somewhat misleading. If the BBay boy represents just one accidental spill over out of many, then host specificity doesn't need to change over time. The results show that *R. felis* was able to infect humans 2000 years ago, and it can still do so today. The only alternative scenario would be that *R. felis* actually became less host specific, but the data couldn't show this either.

In table S2, it is not clear how the number of matching reads are calculated at the genus level. E.g. did they map against all references from a genus, and report de-duplicated reads?

SI9: the text in that section is well-written, but the title of the section is not an accurate description of the information that is provided in the following text.

Reviewer #2 (Remarks to the Author):

The authors reconstructed a historical *Rickettsia felis* genome from metagenomic sequencing of a human remain, and compared the reconstructed genome with modern *Rickettsia* genomes to extrapolate the age of all *R. felis* to 2.5 million years ago. The authors also proposed the presence of *Toxoplasma gondii* as well as at least 7 other major human pathogens in the metagenomic data.

This is an interesting manuscript. The identification and authentication of *R. felis* reads, and the subsequent genome reconstructions are solid and valuable. The other parts of the manuscript, however, need more justifications.

Kraken has been widely used in many metagenomic projects. Its results, however, are known to bear with the problem of high false-positives (Dilthey, Nat Comm, 2019; Hübner Genome Biology, 2019). In particular, Kraken often reports pathogens that are not present in ancient samples (Zhou, RECOMB 2018). It is surprising that the authors have accepted all Kraken predictions without any further verification. In particular, the authors reported that the so called "T. gondii specific" reads have very high duplication rates (>92%), and most of the remaining reads have been mapped to repetitive regions. This is a strong signal of non-specific matches and therefore highly suspicious.

The manuscript also has major faults on the dating of the *Rickettsia* genomes. The authors used a constant prior and strict molecular clock rate for the BEAST analysis. However, it is unrealistic to assume a constant population size across multiple species, nor a constant mutation rate. The authors should at least run TEMPEST before the BEAST run, to see whether there is enough temporal signal (I

don't think so) for their analysis.

Minor points:

Line 36: meta-genome -> metagenome

Line 130: Which reference genome was used to estimate the ~99.90% coverage of aDNA reads?

Line 133 and 134. Why the closest homologue to BBayA is LSU-Lb (ANI 99.90%) rather than URRWxCaI2 (ANI 99.95%) ?

Fig. 2a. It will be much better if a complete genome (URRWxCaI2) can be used as the reference in the figure.

Line 160. What sort of "genotypic and phenotypic attributes" can make a phylogenetic placement be difficult? Do the authors refer to recombination or incomplete lineage sorting?

Line 166-168. Firstly, the sentence refers to Fig 3 rather than Fig. 2. Secondly, the statement is not true, because the authors ignored the whole SFG group in Figure 3, which, according to Fig. s3, sits between *R. typhi* and *R. felis*.

Fig. 3. The branches in the *R. felis* are so short that nobody can see the branching orders.

Line 220-223. Please visualize the read depths along the chromosomes for *T. gondii*.

Line 275-276. This statement is not correct, because any genomic fragment that is present in aDNA but not in the modern reference will not be identified. Therefore we don't know whether there is any major loss of genomic island in modern genomes comparing the ancient genome.

Reviewer #3 (Remarks to the Author):

This study reports the first ancient *Rickettsia felis* genome and genome-wide data for ancient *Toxoplasma gondii*, both recovered from the ~2000 year old skeletal remains of a boy from Ballito Bay, South Africa. This is one of the first studies to produce ancient pathogen genome data from the African continent, which brings not only novel pathogen species to the aDNA field, but also much needed geographical diversity to the current literature. This study is also important because it provides new insight into the past geographical spread of these pathogens, as well as some insight into the evolutionary history of *Rickettsia felis* through molecular dating, genome characterization and investigation of the pangenome. Overall, I think the findings of this paper are of high value, however, the manuscript needs some revision with regard to providing clarity of the methods used and more in-depth interrogation of some of the results.

Major

1) To appeal to the microbe-oriented audience I would like to see some of the information presented in various SI sections brought forward to the introduction. The focus on human evolution is heavy in the intro, and although these are certainly very interesting points and deserve to be mentioned, it would be good if more context were provided in the intro for the two pathogens that are the focus of this study.

1a) Out of curiosity, are there any modern studies that report the co-infection of *R. felis* and *T. gondii*?

2) It would be nice if the authors could investigate, or mention something if it's already been investigated, in the results regarding the presence/absence of *Rickettsia* virulence genes in the ancient *R. felis* genome compared to the modern *Rickettsia*. I appreciate that it is an understudied microbe and that there is little modern data to go on, but it would still be nice to see such an analysis. Have the authors investigated the potential presence of larger deletions? I see in the S4 supplementary tables that indels have already been analysed, but I am wondering if there are any larger deletions, which might point to genes that have been acquired more recently by modern counterparts. Deletion breakpoints can be investigated by mapping with BWA-Mem (see point below regarding bwa-mem vs.

bwa-aln) and counting the number of reads spanning the deletion breakpoints.

2a) Furthermore, when investigating the presence/absence of virulence genes, or generally carrying out competitive mapping the mapping quality (samtools view MAPQ(-q)) should be set to 0 so that reads that map equally well in two places can be mapped. However, when doing competitive mapping to multiple concatenated genomes (which is what I assume has been done here, it should be specified in the methods) where the aim is to investigate which genome has the highest number of unique assigned reads, this should be done with MAPQ 37 so that reads that map equally well to more than one genome are excluded. This method was applied by (Andrades Valtuena, Mittnik et al. 2017), for example, to screen for the presence of *Y. pestis*. The mapping parameters need to be made clear throughout this manuscript.

3) More information needs to be provided regarding the construction of the Maximum Likelihood tree (Figure 3, methods lines 600-607). Based on the tree view provided there appears to be no branch shortening between the BBayA *R. felis* genome and its modern counterparts. If this shows up when the tree is zoomed in, please provide such a view.

3a) details as to what data the tree is based on needs to be provided. Is it based on all sites in the multiple genome SNP alignments? Was partial deletion or complete deletion used? If complete deletion was used, then I suggest the authors experiment with using partial deletion, as this might reveal some branch shortening of the ancient BBayA *R. felis* genome.

3b) How many SNPs are the trees based on. What is the total number of bases in the complete dataset used for phylogenetic analysis and how many were retained in the tree if partial/complete deletion is/was applied?
Is the tree based on a multi-SNP alignments of the concatenated codon alignment, or all sites of the multi-codon alignment?

4) I recommend using RAxML instead of the MegaX for generating maximum likelihood tree, since MegaX is limited in the parameters that you can apply to your data. Can you do a comparison to check that the same topology is achieved with RAxML as with MegaX?

5) Table 1: please show the % of the *R. felis* and *T. gondii* genomes covered 1-fold (and 5-fold for *R. felis*)

6) Please be more specific throughout the manuscript about which mapping tools (bwa/bowtie), parameters and algorithms (bwa-mem or bwa-aln) used for the different analyses, such as the competitive mapping.

6a) Line 581: BWA v0.6.2-r126 is specified but the citation provided is Langmead and Salzberg 2012, which is for the 'Bowtie' mapper. If BWA was used, which algorithm was used? Mem or aln? Mem is not suitable for mapping ancient microbial DNA from metagenomic samples since it does local alignment with soft-clipping, which means that if a region of a read maps to the reference, then that read will be mapped and the non-matching regions around that matching part will be clipped. This allows for a high probability that closely matching reads from related microbes from the burial environment/soil will be mapped. Aln uses semi-global alignment and is much better suited to mapping ancient DNA. The use of mem by the authors to remove the human host DNA is OK (Line 570-571), you might run the risk of potentially removing too much, this is a stringent approach.

6b) quality trimming of 20 is used for adapter removal, but this can trim the 5-prime end significantly, leading to many duplicate reads remaining in the dataset. Consider using the new option '--preserve5p' with AdapterRemoval 2.3.1 (<https://github.com/MikkelSchubert/adapterremoval/issues/32#issuecomment-504758137>)

6c) What was the mapquality (MAPQ) score they used for 'samtools view' for filtering mapped reads when doing the competitive mappings? If MAPQ 37 was used this prevents reads that can map in more than one place from remaining in the alignment, and would be suitable for identifying which *R. felis* or *T. gondii* species is present. If MAPQ of 0 was used then this allows reads that map equally well to more than one place in the concatenated alignment (I assume this is what has been done, could you please specify in the methods?)

7) Mapping statistics for the laboratory negative controls need to be presented as part of the results.

Below are reviews related to the reporting of additional microbial genera as pathogens: *Anaplasma*, *Babesia*, *Bordetella*, *Brucella*, *Leishmania*, *Plasmodium* and *Trypanosoma*

8) I strongly suggest that the authors sequence a soil sample from the cave where the BBayA boy was buried to rule out the presence of the genera *Anaplasma*, *Babesia*, *Bordetella*, *Brucella*, *Leishmania*, *Plasmodium* and *Trypanosoma* as contaminants from the burial environment, as well as rule out *R. felis* and *T. gondii* as originating from the BBayA boy and not the soil. If any of the putative pathogens presented here are ancient contaminants, then one expects them also to have accumulated aDNA of damage patterns.

9) The vast majority of the microbial diversity in the soil has not been genomically characterized and it is immensely feasible that something similar could be causing false positive hits/mapping reads. The authors provide a rundown of the assigned reads based on competitive mapping, but have the authors investigated which parts of the genomes the reads are mapping too? Are they evenly distributed across the genome? Or are they stacking in certain regions? The damage pattern that provided for *Anaplasma* do not look like that of aDNA and *Bordetella* and *Brucella* do have a huge amount of mapping reads with a high number of mismatches, as indicated by the placement of many of the grey lines that represent all other possible base changes.

10) Do they have close relatives in the environment? A close relative to *Brucella* for example is the soil bacterium *Ochrobactrum anthropi* which is also a member of the *Brucellaceae* family (Ermakova 2017).

Bacteria of the *Bordetella* genus are frequently recovered found in soil, water, sediment etc. (Soumana et al 2017).

The *Babesia* parasite, *Leishmania* parasites and *Plasmodium* are all eukaryotic. Despite the removal of the human mapping reads prior to this analysis it does not exclude contamination from eukaryotic DNA deriving from plants, animals (insects) and fungi from the from the burial environment. With such a large amount of sequencing data (4.8 billion reads) it is not surprising that some reads map to these genomes.

Plasmodium has the best aDNA damage pattern and looks like it could be correct.

For the eukaryotic parasites the authors should investigate to see if any of the mtDNA is present in their shotgun data. This would provide some further support to their claim and perhaps some further resolution as to what is present in their data, especially for *Plasmodium*.

11) I would also like to see these findings cross validated with BLAST analysis for the unique reads.

12) The authors should provide the raw kraken results in the supplementary, for both samples and laboratory negative controls.

13) The case fatality rates for the putative positive genera should be removed from the manuscript/table S5. You should not impose fatality rates from modern pathogens onto data where you don't know what the pathogen is, or if it even is a pathogen rather than a soil contamination. This is pure speculation and should be removed.

14) The petrous bone has previously been shown to be a poor source for ancient pathogen DNA compared to teeth, at least for *Y. pestis* which is a blood borne, likely due to the fact that there is restricted blood flow to the cochlea which is the densest part of the human skeleton (Margaryan, Hansen et al. 2018). Please provide more context in the manuscript as to why you find *R. felis* and *T. gondii* in the petrous bone. It would be good if you could mention in the main manuscript more about which parts of the body/organs these pathogens affect and where you might expect to find them? are they blood borne? etc.

Minor

Line 32: Cribra orbitalia is not pathognomonic of anaemia. Here it is stated that this individual was anaemic without any given context or differential diagnosis provided. Just because this boy had Cribra orbitalia, does not mean that he was anaemic. He was 'possibly' anaemic. This is too strong a statement to make in the opening sentence without any context or differential diagnosis provided. See Walker, Bathurst et al. (2009) for example. The authors provide more context in the discussion, lines 255-260. I suggest simply removing the word anaemic from this sentence.

Line 36-37: It is unclear to me what the authors mean here with the sentence starting "These data contain...". Do you mean that the remains potentially contain traces of pathogenic microbes or that they contain microbes that are potentially pathogenic, such as opportunistic pathogens which are common in microbiomes (oral, faecal, skin etc.). This need to be clarified as I suspect the authors mean the former, but it currently reads as if they mean the latter.

Line 45: 'hypothesized' instead of 'presumed' would be better here

Line 60: "... their influence on the biological and socio-cultural evolution of our species in Africa is routinely overlooked...". Can the authors provide some sort of reason for why this might be? Is it due to methods, dearth of available material, poor preservation of available material or lack of interest in the field for this topic?

Line 66-72: The sentence "In addition, the bio-geographic distribution of *Plasmodium falciparum* (Tanabe et al., 2010) and *Helicobacter pylori* (Linz et al., 2007) exhibits declining genetic diversity, with increasing distance from Africa, with 'Out of Africa' estimates of ~58 kyr and ~80 kyr ago, respectively" needs to be backed up with an explicit statement for why exhibiting declining genetic diversity with distance from Africa is relevant to mention here. This information is already stated in SI 2, lines 720-724 and it would be nice to see at least one short sentence about it in the introduction.

Line 242-243: please provide a citation here

Line 284: '*Salmonella enterica*' is too broad here. Key et al. 2020 refers specifically to the Ancient Eurasian Super Branch (AESB), which refers to a small set of *S. enterica* ssp. *enterica* genomes, for which over 2600 serovars exist.

Line 571: version of BWA used should be stated

Line 593: if applicable, please state any specific parameters used when running FastANI and Snippy

Line 1303: typo "the those"

Supplementary: there are two sections labelled SI 7 starting lines 825 and 872

Citations:

Andrades Valtuena, A., A. Mittnik, F. M. Key, W. Haak, R. Allmae, A. Belinskij, M. Daubaras, M. Feldman, R. Jankauskas, I. Jankovic, K. Massy, M. Novak, S. Pfrengle, S. Reinhold, M. Slaus, M. A. Spyrou, A. Szecsenyi-Nagy, M. Torv, S. Hansen, K. I. Bos, P. W. Stockhammer, A. Herbig and J. Krause (2017). "The Stone Age Plague and Its Persistence in Eurasia." *Curr Biol* 27(23): 3683-3691 e3688.

Margaryan, A., H. B. Hansen, S. Rasmussen, M. Sikora, V. Moiseyev, A. Khoklov, A. Epimakhov, L. Yepiskoposyan, A. Kriiska, L. Varul, L. Saag, N. Lynnerup, E. Willerslev and M. E. Allentoft (2018). "Ancient pathogen DNA in human teeth and petrous bones." *Ecol Evol* 8(6): 3534-3542.

Walker, P. L., R. R. Bathurst, R. Richman, T. Gjerdrum and V. A. Andrushko (2009). "The causes of porotic hyperostosis and cribra orbitalia: A reappraisal of the iron-deficiency-anemia hypothesis." *American Journal of Physical Anthropology* 139(2): 109-125.

28 April 2022

Rebuttal Letter

Re: *Rickettsia felis* DNA recovered from a child who lived in southern Africa 2,000 years ago.

Dear Reviewers,

We thank you for your time in providing us with such detailed and constructive reviews concerning the above manuscript following submission to Nature Communications Biology.

Please note that, following substantial revision of the manuscript, and based on the reassessment of previously included microbial taxa and datasets, we now focus exclusively on the description of an ancient *Rickettsia felis* genome, to the exclusion of previously included pathogenic taxa, including *Toxoplasma gondii*, and also the genera *Anaplasma*, *Babesia*, *Bordetella*, *Brucella*, *Leishmania*, *Plasmodium* and *Trypanosoma*. Accordingly, we do not address concerns regarding these taxa, as these are no longer included in the revised manuscript.

In addition, and given a lack of sufficient temporal signals in the ancient DNA dataset, we were unable to determine chronometric stages in the evolution of the BBayA *R. felis*, including the emergence of a most recent common ancestor (MRCA) for the southern African *R. felis* group. Although this section is no longer included in the manuscript, we do provide full details of our attempts to clarify the phylogenetic dating of our ancient *R. felis* strain in both the Supplementary Information (*i.e.*, SI 5) and the Materials and Methods sections. We do not, therefore, address these concerns, in detail.

Finally, and given concerns about the clarity and illustrative relevance of Figure 3, we have included a fully-revised Figure (*i.e.*, Fig. 3) to replace that included in the originally-submitted manuscript. For the sake of further clarity concerning our phylogenomic reconstruction of the *Rickettsia* genus including our ancient *R. felis* strain, we have also included an additional Figure S4 and a Table S5, as indicated in the caption for Figure 3 in the manuscript.

Please see our reply to the commentary provided by the Reviewers, below.

REVIEWER 1:

Major concerns:

1. With regards to the concerns relating to the ‘large amount of unclear language in the paper’, we have addressed the grammar and have attempted to clarify all changes in the manuscript, as indicated in red text in the revised version herewith re-submitted.
2. As stated above, we focus exclusively on the description of the ancient *Rickettsia felis* genome, to the exclusion of previously included pathogenic taxa, including *Toxoplasma gondii*, and also the genera *Anaplasma*, *Babesia*, *Bordetella*, *Brucella*, *Leishmania*, *Plasmodium* and *Trypanosoma*. In this regard, we note that initial taxonomic classification was achieved using Kraken2, and a custom database of bacterial, archaeal, protozoal and viral genomes from the NCBI RefSeq database (<https://www.ncbi.nlm.nih.gov/refseq/>). Thereafter, and to confirm that the organism represented in our metagenomic output was an *R. felis* strain, and not a closely-related southern African species (*e.g.*, *R. prowazekii*, *R. typhi*, *R. conorii* and *R. africae*), and to detect signs of plasmid

rearrangements, we mapped our datasets against all currently available (*i.e.*, 126) NCBI *R. felis* genomes (Table S3).

3. As indicated above, we mapped our datasets against all currently available (*i.e.*, 126) NCBI *R. felis* genomes, with no available genomes being excluded from our analyses.

4. Contrary to our original analyses and results, and given concerns regarding the similarity of our ancient *R. felis* strain to either *R. felis* LSU-Lb or *R. felis* URRWxCal2, we determined that our *R. felis* strain is comparable to both the *R. felis* URRWxCal2 (GCA_000012145.1) and the *R. felis* LSU-Lb (GCA_000804505.1) reference genomes, with an average nucleotide identity (ANI) of 99.95% and 99.90%, respectively (Table S4 a, b, c, d and e). Subsequent phylogenetic analysis revealed that the *R. felis* LSU-Lb strain is the closest homologue to the ancient *R. felis* strain.

5. With regards to the request that the phylogenetic analyses of the ancient *R. felis* strain should be re-considered, we proceeded to re-analyses all our data and generate a fully-revised Figure, in addition to which we have also included a new and additional Figure S4 and a Table S5, as indicated in the caption for Figure 3 in the manuscript.

6. As we focus solely on the presence of *R. felis*, and not on the severity of a possible co-infection with several additional taxa, *i.e.*, *Toxoplasma gondii* and also the genera *Anaplasma*, *Babesia*, *Bordetella*, *Brucella*, *Leishmania*, *Plasmodium* and *Trypanosoma*, the matter of clarifying multiple co-infection as a reason for the demise of the child is no longer relevant to the manuscript and to our conclusions.

Minor concerns:

Line 39: With regards to the completeness of the ancient *R. felis* genome, we now indicate that ‘This genomic structure allows high mapping specificity across the *R. felis* genome, which allowed us to infer the presence and absence of genomic regions from the level of coverage observed after mapping the raw datasets to the reference genomes (Fig. 2a). We were able to recover ~99.90% of the *R. felis* chromosome at a mean depth of coverage of 11.41-fold, and assemble 99.53% of the ancient *R. felis* genome’.

Line 155: Edit distances are not used to confirm antiquity. Edit distances are used mainly to indicate genomic similarities, and we state that ‘Distribution of edit distance of high quality BBayA *R. felis* reads mapping to *R. felis* LSU-Lb and *R. felis* URRWxCal2’.

Lines 275/286: Following revision of the indistinct statement referred to, we now simply conclude that ‘We cannot detect, with certainty, changes in either the virulence or host specificity, over the past ~2,000 years of evolutionary history of *R. felis* in southern Africa’.

REVIEWER 2:

We much appreciate the constructive commentary by Reviewer 2 that ‘This is an interesting manuscript. The identification and authentication of *R. felis* reads, and the subsequent genome reconstructions are solid and valuable’. We also heed this Reviewers concerns, and herewith aim to clarify the issues brought to our attention.

Major concerns:

1. As noted above, we reiterate that initial taxonomic classification was achieved using Kraken2 and a custom database of bacterial, archaeal, protozoal and viral genomes from the NCBI RefSeq database. Thereafter, and to confirm that the organism represented in our metagenomic output was an *R. felis* strain, and not a closely-related southern African species (*e.g.*, *R. prowazekii*, *R. typhi*, *R. conorii* and *R. africae*), and to detect signs of plasmid rearrangements, we mapped our datasets against all currently available (*i.e.*, 126) NCBI *R. felis* genomes (Table S3). Accordingly, the exclusion of the various pathogenic taxa formerly included in the manuscript renders this concern inconsequential, as we now focus solely on the identification and analyses of *R. felis*.

Minor concerns:

Lines 36, 130 and 133: With regards to the use of modern reference genomes used against which to map our ancient *R. felis* dataset, and the level of similarity detected, we state that ‘The BBayA dataset contains an *R. felis* strain comparable to both the *R. felis* URRWxCal2 (GCA_000012145.1) and the *R. felis* LSU-Lb (GCA_000804505.1) reference genomes, with an average nucleotide identity (ANI) of 99.95% and 99.90%, respectively (Table S4 a, b, c, d and e)’. We no longer emphasise the erroneous assumption that the ancient *R. felis* genome has *R. felis* LSU-Lb (GCA_000804505.1) as its closest homologue, as opposed to *R. felis* URRWxCal2, at 99.95%. In addition, we further note that ‘The *R. felis* LSU-LB and URRWXcal2 strains were used as reference genomes during the BWA v0.6.2-r126 alignment to the BBayA *R. felis* chromosomes and plasmids’.

Line 166: The branches for *R. felis* are indeed short, and these do make it near-impossible to discern any clear branching between the species. There is, unfortunately, little to be done to improve this, besides including an inset (Fig. 3c) in which the branching region is enlarged for the reader’s convenience. As stated above, and for the sake of clarity concerning our phylogenomic reconstruction of the *Rickettsia* genus including our ancient *R. felis* strain, we have included an additional Figure S4 and a Table S5, as referred to in the caption for Figure 3 in the manuscript.

REVIEWER 3:

We appreciate the commentary by Reviewer 3 that ‘This is one of the first studies to produce ancient pathogen genome data from the African continent, which brings not only novel pathogen species to the aDNA field, but also much needed geographical diversity to the current literature. This study is also important because it provides new insight into the past geographical spread of these pathogens, as well as some insight into the evolutionary history of *Rickettsia felis* through molecular dating, genome characterization and investigation of the pangenome’.

We acknowledge the comment that ‘Overall, I think the findings of this paper are of high value, however, the manuscript needs some revision with regard to providing clarity of the methods used and more in-depth interrogation of some of the results’ and have addressed all the concerns raised by this Reviewer.

Major concerns:

1. As per the request of Reviewer 3, we have, to appeal to the microbe-oriented audience, included a large amount of the information presented in various SI sections within the manuscript, as indicated in red text.
2. We do address the presence/absence of *Rickettsia* virulence genes in both the ancient *R. felis* genome compared to the modern *Rickettsia*, stating that ‘With regards to the pathogenicity of the BBayA *R. felis* strain, the presence of both the Sca2 (pRF25) and the RHS-like toxin (pLbaR-38) mutations suggests that this ancient strain was, in all probability, just as pathogenic as contemporary *R. felis* variants, and that it may well have resulted in symptoms typical of typhus-like flea-borne rickettsioses, including fever, fatigue, headache, maculopapular rash, sub-acute meningitis and pneumonia (SI 7)’.
3. To confirm that the organism represented in our metagenomic output was an *R. felis* strain, and not a closely-related southern African species (e.g., *R. prowazekii*, *R. typhi*, *R. conorii* and *R. africae*), and to detect signs of plasmid rearrangements, we mapped our datasets against all currently available (*i.e.*, 126) NCBI *R. felis* genomes (Table S3). For performing competitive mapping using multiple concatenated genomes, the mapping quality (samtools view MAPQ(-q)) was set at MAPQ 37 to ensure that any reads that map equally well to more than one genome were excluded.

In addition, and as the branches for *R. felis* are indeed short, making it near-impossible to discern clear branching between the species, we do include an enlarged inset (Fig. 3c) in which the branching region is more apparent. As stated above, and for the sake of further clarity concerning our phylogenomic reconstruction of the *Rickettsia* genus

including our ancient *R. felis* strain, we have included an additional Figure S4 and a Table S5, as referred to in the caption for Figure 3 in the manuscript.

4. Details as to what data the tree is based on are clearly provided in the Materials and Methods section. Briefly, we used a concatenated codon alignment produced from 138 protein sequences from 127 genomes, *i.e.*, the 126 available *Rickettsia* genomes and also our ancient *R. felis* genome.

5. As per the recommendation of Reviewer 3, we have employed RAxML instead of the MegaX for generating maximum likelihood tree. Additional information is provided in Figure S4, with the caption reading ‘Additional maximum-likelihood trees constructed using the same codon alignment with the FastTree version 2.1.10 with -gtr and -gamma options, the RAxML version 8.2.11 with -m GTRGAMMA, -#100 (to search the best tree between 100 replicates) and -# autoMR option to determine node bootstraps with automatic number of replicates and MEGA-CC version 11.0.10 using GTR (G+I) model and bootstrap support of 100 replicates. These phylogenomic reconstructions were compared using the approximately unbiased (AU) test implemented in IQ-TREE v.1.5.5 with the options -n 0 -zb 10000 -au -zw. The *p*-values for the AU test of the FastTree (*p*-value 0.337), IQtree (*p*-value 0.727) and RAxML (*p*-value 0.315) reconstructions indicated these trees as 95 % confident sets, while the MEGA-CC tree got a significant exclusion (*p*-value 0.000127)’.

6. As per the request of Reviewer 3, we state, in the Materials and Methods section, that ‘Paired-end aDNA sequencing reads were first processed to facilitate the removal of adapters and primers using AdapterRemoval v2654 following the parameters ‘min-quality’ 20, ‘min-length’ 35 and ‘collapsed to merge’ the forward- and reverse-sequence reads. Human (*i.e.*, *H. sapiens*) reads were removed using the BWA-MEM algorithm against the human reference genome⁶⁵. Using the new option ‘--preserve5p’ with AdapterRemoval 2.3.1 (<https://github.com/MikkelSchubert/adapterremoval/issues/32#issuecomment-504758137>) resulted in a comparable DNA damage plot (Fig. S3)’.

7. As indicated above, ‘Paired-end aDNA sequencing reads were first processed to facilitate the removal of adapters and primers using AdapterRemoval v2654 following the parameters ‘min-quality’ 20, ‘min-length’ 35 and ‘collapsed to merge’ the forward- and reverse-sequence reads’.

8. With regards to the inclusion of data derived from the sequencing and analyses of negative controls, we aim to clarify that the originally-published study did not detect any taxa in the DNA extraction controls, and that these were, accordingly, not sequenced’. We further state that ‘DNA was extracted from between 60 and 190 mg of bone powder using silica-based protocols with modifications, and were eluted in 50-110 µl Elution Buffer (Qiagen). Between 3 and 6 DNA extracts were made for each individual (or accession number) and one negative extraction control was processed for every 4 to 7 samples extracted. The optimal number of PCR cycles to use for each library was determined using quantitative PCR (qPCR) in order to see at what cycle a library reached the plateau (where it is saturated) and then deducting three cycles from that value. The 25 µl qPCR reactions were set up in duplicates and contained 1 µl of DNA library, 1X Maxima SYBR Green Mastermix and 200 nM of each IS7 and IS8 primers and were amplified according to supplier instructions (ThermoFisher Scientific). Each library was then amplified in four or eight reactions using between 12 and 21 PCR cycles. One negative PCR control was set up for every four reactions. Blunt-end reactions were prepared and amplified using IS4 and index primers. Damage-repair reactions had a final volume of 25 µl and contained 4 µl DNA library and the following in final concentrations; 1X AccuPrime Pfx Reaction Mix, 1.25U AccuPrime DNA Polymerase (ThermoFisher Scientific) and 400nM of each the IS4 and index primers. Thermal cycling conditions were as recommended by ThermoFisher with an annealing temperature of 60°C⁶⁶. The resulting libraries were quantified either on a TapeStation using a High Sensitivity kit (Agilent Technologies) or using a Bioanalyzer 2100 and a High Sensitivity DNA chip (Agilent Technologies). We further state that ‘Regrettably, and given that the DNA extraction controls did not yield any DNA, and were therefore not sequenced², it is not possible to include any information regarding the analyses of taxa detected in negative controls in this study, although this is standard practice in aDNA-related research’.

9. With regards to the request for cross-validation of our results via BLAST analysis, we note that we focus exclusively on the description of the ancient *Rickettsia felis* genome, to the exclusion of previously included

pathogenic taxa, including *Toxoplasma gondii*, and also the genera *Anaplasma*, *Babesia*, *Bordetella*, *Brucella*, *Leishmania*, *Plasmodium* and *Trypanosoma*. In this regard, we note that initial taxonomic classification was achieved using Kraken2, and a custom database of bacterial, archaeal, protozoal and viral genomes from the NCBI RefSeq database (<https://www.ncbi.nlm.nih.gov/refseq/>). Thereafter, and to confirm that the organism represented in our metagenomic output was an *R. felis* strain, and not a closely-related southern African species (*e.g.*, *R. prowazekii*, *R. typhi*, *R. conorii* and *R. africae*), and to detect signs of plasmid rearrangements, we mapped our datasets against all currently available (*i.e.*, 126) NCBI *R. felis* genomes (Table S3). As indicated above, we mapped our datasets against all currently available (*i.e.*, 126) NCBI *R. felis* genomes, with no available genomes being excluded from our analyses.

10. As noted above, and with regards to the inclusion of data derived from the sequencing and analyses of negative controls, we aim to clarify that the originally-published study did not detect any taxa in the DNA extraction controls, and that these were, accordingly, not sequenced’.

We furthermore do address the concerns about petrous bone having previously been shown to be a poor source for ancient pathogen DNA compared to teeth, citing the study by Margaryan et al. (2018). We provide additional context in the manuscript concerning this matter, via the following addition to the Discussion section of the revised manuscript: ‘Since the DNA analysed in this study was extracted from bone (petrous, *i.e.*, LPB and RPB) and tooth (upper premolar, *i.e.*, ULPM) samples, the fact that not all pathogenic microbial taxonomic categories might be recoverable from either human skeletal or dental remains⁴¹ suggest that some taxa might be underrepresented. In this regard, our study confirms that the DNA of ancient pathogenic microbial taxa can be recovered from human petrous bone. A previous study⁴¹ focussed on the differential detection of a single pathogen, *Yersinia pestis*, from teeth and petrous samples, showing a higher microbial diversity in teeth than petrous bones, including additional pathogenic and oral taxa. The reasons cited for this result include the fact that the otic capsules of the petrous bones are harder than tooth roots, implying that very little exogenous DNA will penetrate into these bones. Differences in blood circulation and bone turnover rates in petrous regions and teeth may therefore account for the variable incidence of pathogenic DNA in these respective sample sites. In this instance, the remains analysed represented that of a child, the skull and teeth of which were still experiencing formative development and, therefore, not yet fully fused, developed and densified. In addition, chronic diseases and resulting comorbidities are associated with diminished bone mineral accrual and bone loss, and various paediatric disorders have been implicated in impaired bone health⁴². It is therefore probable that the child displayed irregular and abnormally-low skeletal bone density and skeletal metabolism, in turn resulting in an increasing predisposition of pathogenic microbes to circulate through and enter the dense otic capsules of the petrous regions. Consequently, this resulted in the extraction of >99% of ancient *R. felis* reads analysed in this study, from the RPB and LPB samples. In addition to the young age and compromised health of the child, taphonomic factors might further explain the differential preservation of microbial DNA in the petrous and tooth (ULPM) samples. The context from which the child’s remains were retrieved comprised shell midden overlooking the beach, ~ 46 m from the high-Indian Ocean water mark. The humid saline conditions and loose sedimentary matrix may certainly have resulted in increasingly rapid DNA degradation, particularly in the exposed sub-adult teeth⁴⁷. Ultimately, the sequencing strategy originally employed resulted in the sequencing of seven libraries derived from the left (LPB) and right (RPB) petrous samples, and only a single library from the upper-left premolar (ULPM), introducing a bias in terms of the numbers of microbial reads recovered from the respective samples’.

Minor concerns:

Line 32: We have addressed the occurrence of cribra orbitalia as not necessarily pathognomonic of anaemia, stating that ‘Indications of cribra orbitalia are a symptom of marrow expansion caused by haemopoietic factors, and has been attributed to both malnutrition (*e.g.*, megaloblastic anaemia) and parasitism. However, other plausible causes for this pathology include malaria (*Plasmodium* sp.), hookworm infection (*Ancylostoma duodenale* and *Necator americanus*) and schistosomiasis (*Schistosoma haematobium*), the latter of which was suggested as the best-fit cause for the child’s pathology. Besides *R. felis*, which cause comparable osteological pathologies, the pathogens referred to above were absent from our dataset. *R. felis* is an obligate intracellular pathogen which, in order to

establish productive infections, also modifies the cytoskeletal architecture and the endomembrane system of their host cells (SI 7)'.

Line 36: We state that 'During the generation of shotgun metagenomic DNA sequence data from human skeletal material, large amounts of associated genetic data are generated. These data frequently contain traces of pathogenic microbes associated with the person whose DNA was examined', implying that ancient human DNA datasets often do contain traces of pathogenic microbes.

Line 60: We clarify the reasons for our statement stating that 'Despite the fact that pathogens have long exerted a significant influence on hominin longevity and human genetic diversity, and given that diseases continue to shape our history, their influence on the biological and socio-cultural evolution of our species in Africa is routinely overlooked, perhaps because of a combination of limited available material and poor preservation of DNA, resulting in lack of interest in the topic in Africa (SI 2)'.

Line 66: To clarify the significance of ancient associations between humans and certain pathogens, we now state that 'The presence of specific TLR4 polymorphisms (i.e., pathogen-recognition receptors) in African, as well as in Basque and Indo-European populations, suggests that some mutations arose in Africa prior to the dispersal of *H. sapiens* to Eurasia. In addition, the bio-geographic distribution of *Plasmodium falciparum* and *Helicobacter pylori* exhibits declining genetic diversity, with increasing distance from Africa mirroring past human expansions and migrations, with 'Out of Africa' estimates of ~58 kyr and ~80 kya, respectively. Given the long association of *H. pylori* with humans, the current population structure of *H. pylori* has been regarded as mirroring past human expansions and migrations'.

Line 571: With regards to the use of BWA, we state that 'The *R. felis* LSU-LB and URRWXcal2 strains were used as reference genomes during the BWA v0.6.2-r126 alignment to the BBayA *R. felis* chromosomes and plasmids', and also that 'Paired-end aDNA sequencing reads were first processed to facilitate the removal of adapters and primers using AdapterRemoval v2654 following the parameters 'min-quality' 20, 'min-length' 35 and 'collapsed to merge' the forward- and reverse-sequence reads. Human (i.e., *H. sapiens*) reads were removed using the BWA-MEM algorithm against the human reference genome'.

Line 593: With regards to the parameters used whilst employing FastANI and Snippy, we state that 'The assembled ancient *R. felis* genome was used for average nucleotide identity and single nucleotide variant analysis using the FastANI68 and Snippy (<https://github.com/tseemann/snippy>) software programmes at default parameter settings'. The assembled ancient *R. felis* genome was used for average nucleotide identity and single nucleotide variant analysis using the FastANI68 and Snippy (<https://github.com/tseemann/snippy>) software programmes, respectively.

Reviewers' comments:

Reviewer #1 (Remarks to the Author):

Review for

Rickettsia felis DNA recovered from a child who lived in southern Africa 2,000 years ago

I appreciate the authors comments and revisions of this manuscript. The authors have adequately addressed concerns regarding the absence of a temporal signal for molecular dating analyses.

I still have some concerns regarding the identification of *R. felis* reads. The authors state that to confirm that the organism in their metagenomic output was an *R. felis* strain and not a closely related southern African species (e.g. *R. prowazekii*, *R. typhi*, *R. coronii* and *R. africae*), and to detect signs of plasmid rearrangements, they mapped the dataset against all currently available 126 *R. felis* genomes. However, mapping against just *R. felis* genomes does not address the first concern regarding potential closer similarity to related southern African rickettsia species. In order to exclude such similarity, the reads need to be mapped against those species as well.

For the genome reconstruction, the text suggests that reads mapping equally well against multiple genomes were excluded before extraction of FASTQ reads for assembly. Wouldn't this exclude reads that are potentially useful from the assembly?

In the phylogenetic analysis, the authors include 4 *R. felis* genomes. It is not clear how those 4 genomes were chosen, since earlier, 126 genomes were used for genome reconstruction. This may affect further interpretation of the phylogenetic tree. The authors point out the close relationship of the ancient *R. felis* to TG rickettsia in the tree. However, the longer branch length of the TG clade would suggest a closer relationship with the SFG clade.

Lines 218-245 of the discussion is almost identical to SI 6. Regarding the discussion of recovery of pathogen DNA from petrous bone, the authors may wish to note that HBV and VARV reads have been successfully recovered from petrous bone (e.g. Kocher et al., 2021, Muhlemann et al., 2020).

Lines 296-307 are very speculative. There is no evidence from a single genome for or against more frequent instances of *R. felis* infections during the Later Stone Age.

Minor points:

- SI2 is well-written, but it doesn't support the point made on line 54 of the introduction (lack of interest of pathogens on human evolution in Africa).
- Line 60ff: both kyr and kya abbreviations are used.
- Line 160-162: the authors state 'Because *R. felis* displays genotypic and phenotypic attributes of both spotted fever (SFG) and typhus group (TG) rickettsiae, it is difficult to place phylogenetically (Fig. S2)'. I don't see how this follows from figure S2. What would make *R. felis* difficult to place were e.g. recombination.
- The coloring of the *R. helvetica* sequence is different in the different panels of Figure S4 and Figure 3.

Reviewer #3 (Remarks to the Author):

The manuscript is much improved and the increased focus on the *R. felis* genome and *R. felis* pathology has made this a more streamlined read. I enjoyed reading it. My are comments below:

Major revisions:

- The *R. felis* genome presented here is not the first ancient pathogen from the African continent to be studied. See Neukamm 2020 (<https://bmcbiol.biomedcentral.com/articles/10.1186/s12915-020-00839-8>), where they recovered a 2200-year-old *Mycobacterium leprae* strain and a 2000-year-old HBV strain. This needs to be amended several places in the manuscript, e.g. in line 214-216, as it has already been demonstrated that ancient DNA can be recovered from 2000 year old African remains.

-Table S3: In this table the GCF numbers and relevant metadata for accessing these strains should be added. For example, the host type that it was isolated from, human/type of animal etc. The NCBI accession codes should also be listed for each strain used.

- Discussion (main text), lines 218-245: Could you make some reference as to whether *R. felis* is blood-borne? Does it cause sepsis? If it causes sepsis, then you are more likely to find it in diverse parts of the skeleton, versus whether it only causes localised disease. Is there anything in its biology/ the way it causes disease that makes it likely to find it in the petrous bone? You have already done a good job of explaining most of the aspects of why you might find it in the petrous bone, but it would be good to have an additional sentence referring to this directly. In S7 line 188 it is written "*R. felis* has furthermore been detected in the blood and cerebrospinal fluid .." Could its presence in cerebrospinal fluid make it more likely to be found in the petrous bone?

Line 404-417, Methods: please provide further details regarding the de novo assembly, eg. how many contigs did the assembly produce? What was the quality/metrics of those contigs? How were they mapped back to the reference genome/integrated into one genome/sequence to be compared with the modern references? What was the logic for excluding reads that map equally well to more than one *R. felis* genome? These reads are likely from conserved regions among the *R. felis* genomes and would potentially be beneficial for phylogenetic analyses if there is only one *R. felis* strain suspected to be present, which seems to be the case. This might also affect your pangenome analysis and core gene analysis if you have excluded common core genes by filtering them out through exclusion. This also relates to my comment regarding line 406.

Minor revisions:

Line 21: I suggest changing "insufficient" to "little, or no" or something along those lines. Insufficient in this context is vague and broad.

Line 24-26: I suggest making this sentence more assertive, e.g. "We identified ancient DNA sequence reads [...], and reconstructed a ~29X ancient *R. felis* genome". It reads a bit passive as it stands.

Line 33: "... large amounts of associated genetic data are generated", this is a bit vague as a lot of non-human-associated environmental DNA are also generated when using a SG sequencing approach from human remains buried in the soil.

Line 33-34: I suggest replacing "frequently contain" with "may contain"

Line 46-49: This is somewhat repetitive of what is written in line 30-32 and does not mesh well with the rest of that paragraph, I suggest integrating it into the first paragraph of the introduction and beginning third paragraph with something like: "Pathogens have long exerted ..."

Line 52-54: I suggest replacing "routinely overlooked" with "understudied", as the former sounds intentional, when in fact there is great interest in studying ancient pathogens from Africa, but as you state the "limited availability of material and poor preservation of DNA" impacts the possibilities to work on material from African. I also suggest you rephrase "lack of interest" in line 54.

Line 81: Please specify the date the RefSeq database was downloaded for specificity and accuracy, as RefSeq is continually updated.

Line 111: I suggest changing "the child" to the sample/individual's name "BBayA"

Line 116-117 + 136 + elsewhere: For clarity, when you use the word "assembly" it would be beneficial if stated that you are specifically referring to "de novo assembly" in the main text, "de novo" is only mentioned in the methods.

Line 166: suggest replacing "has for" with "its"

Line 188: can you clarify what is meant by a "distinct site pattern"?

Line 178/Figure 3:

These phylogenetic ML trees would be easier to interpret if the GCF numbers of the different *Rickettsia* strains were removed. The GCF number should instead be listed in Table S3, see above comment.

Line 207-209: It would be worth noting here, or at least being aware of, that only a very few pathogenic infections cause distinct skeletal lesions in humans, and only when the person survives with the disease/pathogen for a prolonged period, sometimes decades. These include TB, leprosy, brucellosis and treponemal diseases.

Line 221-222: replace "ancient pathogenic microbial taxa" with "*R. felis*", just because an individual is infected with a pathogen does not mean that you would expect to be able to isolate it from anywhere in the skeleton.

Line 256-2567: The words used here are very specific medical terms. Is there any way of rephrasing it to make it easier to understand for the average reader?

Line 321-330, Materials and Methods: The first section could be divided into two. One for permits/provenience and one for DNA extraction to make it easier to read.

Line 346: clarity needed regarding "...each individual (or accession number)", did you study more than one individual/skeleton? What type of accession number is this referring to? Museum accession?

Line 406: please specify which genomes were used when referring to "multiple concatenated genomes"

22 July 2022

Reviewer Reply Letter

Re: *Rickettsia felis* DNA recovered from a child who lived in southern Africa 2,000 years ago.

Dear Editor,

We thank you for your time in providing us with such detailed and constructive reviews concerning the above manuscript following submission to Nature Communications Biology

Please see our reply to the commentary provided by the Reviewers, below.

Reviewer 1:

*I still have some concerns regarding the identification of *R. felis* reads. The authors state that to confirm that the organism in their metagenomic output was an *R. felis* strain and not a closely related southern African species (e.g. *R. prowazekii*, *R. typhi*, *R. coronii* and *R. africae*), and to detect signs of plasmid rearrangements, they mapped the dataset against all currently available 126 *R. felis* genomes. However, mapping against just *R. felis* genomes does not address the first concern regarding potential closer similarity to related southern African rickettsia species. In order to exclude such similarity, the reads need to be mapped against those species as well.*

We thank the reviewer for the comment as it indicates that we were perhaps not clear in the description of our analyses. Indeed, only four of the 126 reference genomes we used are *R. felis* species, namely, GCF_000804505.1, GCF_000964665.1, GCF_000804525.1, and GCF_000012145.1. The remaining *Rickettsia* genomes used were distributed amongst 32 different species. The reads that were used to assemble the *R. felis* BAA001 genome described in the study were mapped against the obtained assembly, as well the other 126 *Rickettsia* genomes, using the coverm v0.6.1 software (<https://github.com/wwood/CoverM>) with the BWA-MEM mapping tool and with several metrics, including the following: a) In a competitive mapping, the 5 *R. felis* genomes recruited over 80 % of the reads, b) The percentage of reads used for the assembly were over 89 % for the *R. felis* genomes, while numbers below 72 % were found for the other species and c) The percentage of bases of the genome covered for at least 1 read at 95% identity and 80 % of read alignment, was over 98 % for the *R. felis* genomes (excepting the *R. felis* str. Pedreira with 94 %, GCF_000964665.1); while for other species it was below 80 % (GCF_000828125.2, *Rickettsia asemonensis*).

This clarification is also now indicated in the text, as we state that ‘Initial taxonomic classification was achieved using Kraken228 and a custom database of bacterial, archaeal, protozoal and viral genomes from the NCBI RefSeq database (<https://www.ncbi.nlm.nih.gov/refseq/>). This was followed by mapping our dataset against all the currently available (*i.e.*, 126) NCBI *Rickettsia* reference genomes, four of which comprise available *R. felis* genomes, using SAMtools view (<https://github.com/SAMtools/SAMtools>), with the mapping quality (MAPQ(-q)) set at 37 to ensure that any reads that map equally well to more than one genome, were excluded from further analyses’.

For the genome reconstruction, the text suggests that reads mapping equally well against multiple genomes were excluded before extraction of FASTQ reads for assembly. Wouldn't this exclude reads that are potentially useful from the assembly?

We do not understand this comment, especially as the *R. felis* BBayA genome we retrieved was almost complete (99.53%). Furthermore, we used this restrictive approach to avoid recruiting contaminating sequences. This was rather successful to our point of view as it only presented a 0.47% contamination. Furthermore, the quality of all the *Rickettsia* genomes was evaluated with CheckM v1.1.3 software, and include this additional information in the text of the manuscript as Table S5.

In the phylogenetic analysis, the authors include 4 R. felis genomes. It is not clear how those 4 genomes were chosen, since earlier, 126 genomes were used for genome reconstruction. This may affect further interpretation of the phylogenetic tree. The authors point out the close relationship of the ancient R. felis to TG rickettsia in the tree. However, the longer branch length of the TG clade would suggest a closer relationship with the SFG clade.

As previously mentioned, we used the four available *R. felis* genomes, with the others belonging to 32 different *Rickettsia* species. Regarding the comment about the closeness of *R. felis* to TG vs SFG, we performed a clustering analysis based on MASH and FastANI values using dRep v3.2.2. The analyses showed that the TRG clade (into which *R. felis* is classified) is closer to the SFG clade than the TG clade. Notably, the MASH average nucleotide identity values of TRG were ~92 % with SFG and ~ 87 % with the TG group. Therefore, although in the phylogenomic analyses the TRG group closely relates with the root of the TG clade, their genomes are more similar to the SFG clade genomes.

Minor points:

SI2 is well-written, but it doesn't support the point made on line 54 of the introduction (lack of interest of pathogens on human evolution in Africa).

The reference to SI 2 was deleted from Line 53 as it does indeed relate more to the paragraph from Line 60, where it is now referred to.

Line 60: both kyr and kya abbreviations are used.

This was corrected to use only 'kya' in the text.

Line 160-162: the authors state 'Because R. felis displays genotypic and phenotypic attributes of both spotted fever (SFG) and typhus group (TG) rickettsiae, it is difficult to place phylogenetically (Fig. S2)'. I don't see how this follows from figure S2. What would make R. felis difficult to place were e.g. recombination.

We have deleted this sentence in the revised manuscript as *R. felis* clearly is part of the TRG group in all the phylogenetic trees (Fig 3 and Fig S4).

The colouring of the R. helvetica sequence is different in the different panels of Figure S4 and Figure 3.

We have modified the figures accordingly to clarify the placement of *R. felis* and of *R. helvetica*.

Reviewer 3:

The manuscript is much improved and the increased focus on the R. felis genome and R. felis pathology has made this a more streamlined read. I enjoyed reading it. My comments below:

Major revisions:

The R. felis genome presented here is not the first ancient pathogen from the African continent to be studied. See Neukamm 2020 (<https://bmcbiol.biomedcentral.com/articles/10.1186/s12915-020-00839-8>), where they recovered a 2200-year-old Mycobacterium leprae strain and a 2000-year-old HBV strain. This needs to be amended several places in the manuscript, e.g. in line 214-216, as it has already been demonstrated that ancient DNA can be recovered from 2000-year-old African remains.

We do acknowledge the work in North Africa, but as the geographic context of our research falls within sub-Saharan Africa, we indicate that ‘We furthermore demonstrate that the DNA of ancient pathogenic microbial taxa can also be recovered from prehistoric sub-Saharan African human skeletal remains, and from human petrous bone samples (SI 6)’.

Table S3: In this table the GCF numbers and relevant metadata for accessing these strains should be added. For example, the host type that it was isolated from, human/type of animal etc. The NCBI accession codes should also be listed for each strain used.

Table S3 was updated as requested. It now includes the RefSeq accession numbers, as well other information about the various strains. We included their hosts as well as their location and collection dates. These were also used to determine the absence of any temporal signal. Furthermore, the quality of all the *Rickettsia* genomes was evaluated with CheckM v1.1.3 software, as also now indicated in the additional Table S5.

Discussion (main text), lines 218-245: Could you make some reference as to whether R. felis is blood-borne? Does it cause sepsis? If it causes sepsis, then you are more likely to find it in diverse parts of the skeleton, versus whether it only causes localised disease. Is there anything in its biology/ the way it causes disease that makes it likely to find it in the petrous bone? You have already done a good job of explaining most of the aspects of why you might find it in the petrous bone, but it would be good to have an additional sentence referring to this directly. In S7 line 188 it is written “R. felis has furthermore been detected in the blood and cerebrospinal fluid.” Could its presence in cerebrospinal fluid make it more likely to be found in the petrous bone?

We indicate that ‘Differences in blood circulation and bone turnover rates in petrous regions and teeth may therefore account for the variable incidence of pathogenic DNA in these respective sample sites. *R. felis* is blood-borne and has been detected in the blood and cerebrospinal fluid of individuals diagnosed with malaria, cryptococcal meningitis and also scrub typhus (see SI 7)’. We could not establish whether *R. felis* does in fact cause sepsis, and cannot therefore expand on the question concerning this matter. As it is blood-borne, this might relate to the incidence of this pathogen in petrous bone.

Line 404-417, Methods: please provide further details regarding the de novo assembly, e.g. how many contigs did the assembly produce? What was the quality/metrics of those contigs? How were they mapped back to the reference genome/integrated into one genome/sequence to be compared with the modern references? What was the logic for excluding reads that map equally well to more than one R. felis genome? These reads are likely from conserved regions among the R. felis genomes and would potentially be beneficial for phylogenetic analyses if there is only one R. felis strain suspected to be present, which seems to be the case. This might also affect your pangenome analysis and core gene analysis if you have excluded common core genes by filtering them out through exclusion. This also relates to my comment regarding line 406.

With regards to clarifying the above, we include the following addition to the manuscript in order to clarify the analyses: ‘Of the available NCBI RefSeq genomes utilised in our analyses, some displayed lower completeness values (*e.g.*, 90.97 % for GCF_000964995.1) and higher contamination values (*e.g.*, 7.04 % for GCF_000696365.1), as is indicated in Table S3. When the reads were mapped back to the reference genomes, the reads that were used to assemble the *R. felis* genome were mapped against the obtained assembly, as well the other 126 Rickettsia genomes, using Coverm v0.6.1 (<https://github.com/wwood/CoverM>) with the BWA-MEM mapping tool, resulting in a) a competitive mapping with the five *R. felis* genomes recruiting over 80% of reads, b) the percentage of reads used for the assembly were > 89% for the *R. felis* genomes, while numbers < 72% were found for the other species, and c) the percentage of bases of the genome covered for at least 1 read at 95% identity and 80% of read alignment, was > 98% for the *R. felis* genomes, excepting the *R. felis* str. Pedreira with 94%, GCF_000964665.1, while for other species it was < 80% (*e.g.*, GCF_000828125.2, *R. asembonensis*). The obtained *R. felis* BBayA genome consists of 1,543,518 bases and 111 contigs with a N50 of 42,410 bases and the longest contig comprising 121,989 bases. It exhibits a GC value of 32.5% and a coding density of 84. % for 1,613 predicted proteins. The completeness was 99.53%, and contamination 0.47%. Despite the fact that some reads were excluded, the obtained *R. felis* BBayA genome has a completeness of almost 100% (99.53%) and a contamination of only 0.47%. From the other *R. felis* strain genomes, only those from the strains Pedreira and URRWXCal2 achieved 100% completeness, with 0% and 0.47% contamination, respectively. The remaining two *R. felis* strains, namely LSU-Lb and LSU, presented lower completeness and contamination values (at 97.47% and 1.42%, and 94.14% and 0.71%, respectively (Table S3)’.

Minor revisions:

Line 21: I suggest changing “insufficient” to “little, or no” or something along those lines. Insufficient in this context is vague and broad.

This has been changed, as requested, and is indicated in red text in the revised manuscript.

Line 33-34: I suggest replacing “frequently contain” with “may contain”.

This has been changed, as requested.

Line 46-49: This is somewhat repetitive of what is written in line 30-32 and does not mesh well with the rest of that paragraph, I suggest integrating it into the first paragraph of the introduction and beginning third paragraph with something like: “Pathogens have long exerted ...”

This has been changed, as requested.

Line 52-54: I suggest replacing “routinely overlooked” with “understudied”, as the former sounds intentional, when in fact there is great interest in studying ancient pathogens from Africa, but as you state the “limited availability of material and poor preservation of DNA” impacts the possibilities to work on material from African. I also suggest you rephrase “lack of interest” in line 54.

This has been changed, as requested.

Line 81: Please specify the date the RefSeq database was downloaded for specificity and accuracy, as RefSeq is continually updated.

The RefSeq database was downloaded on 14 March 2018.

Line 111: I suggest changing “the child” to the sample/individual’s name “BBayA”.

This has been changed, as requested.

Line 116-117 + 136 + elsewhere: For clarity, when you use the word “assembly” it would be beneficial if stated that you are specifically referring to “de novo assembly” in the main text, “de novo” is only mentioned in the methods.

This has been clarified in the text to read ‘Consistent with the characteristics of aDNA, we detected significant DNA damage patterns for the reads mapping to the *R. felis de-novo* genome assembly (SI 4)’.

Line 166: suggest replacing “has for” with “its”.

This has been changed as requested, and we now indicate that ‘Specifically, the ancient BBayA *R. felis* genome clustered with *R. felis* LSU-Lb within the transitional group (TRG) rickettsiae and its closest relative, *R. felis* LSU-Lb.’

Line 188: can you clarify what is meant by a “distinct site pattern”?

As we refer to the compositional alignment of the nucleotides in the *R. felis* genome, we have changed the sentence to read as follows: ‘The parsimony informative sites were 33,343 nucleotide sites which present 17,683 distinct nucleotide composition patterns’.

Line 178/Figure 3: These phylogenetic ML trees would be easier to interpret if the GCF numbers of the different Rickettsia strains were removed. The GCF number should instead be listed in Table S3, see above comment.

We note that some of the *Rickettsia* genomes do not have strain names and/or have been sequenced more than once. Therefore, we believe that the NCBI RefSeq assembly accession numbers are the best to clearly identify each genome in the figures. We have nevertheless, as recommended, added the GCF numbers of each genome in Table S3.

Line 207-209: It would be worth noting here, or at least being aware of, that only a very few pathogenic infections cause distinct skeletal lesions in humans, and only when the person survives with the disease/pathogen for a prolonged period, sometimes decades. These include TB, leprosy, brucellosis and treponemal diseases.

We address and acknowledge this by the following amended sentence in the text: ‘Formerly, the identification of skeletal pathologies presented the only means by which information concerning the incidence of diseases in the archaeological record could be gained. However, morphological analyses are limited as not all pathogenic infections necessarily result in diagnostic skeletal lesions³⁴’.

Line 221-222: replace “ancient pathogenic microbial taxa” with “R. felis”, just because an individual is infected with a pathogen does not mean that you would expect to be able to isolate it from anywhere in the skeleton.

This has been changed as requested.

Line 256-267: The words used here are very specific medical terms. Is there any way of rephrasing it to make

it easier to understand for the average reader?

Out text section, which reads as follows, necessitates the inclusion of various specific terminologies and is further clarified in SI 7: ‘With regards to the pathogenicity of the BBayA *R. felis* strain, the presence of both the Sca2 (*pRF25*) and the RHS-like toxin (*pLbaR-38*) mutations suggests that this ancient strain was, in all probability, just as pathogenic as contemporary *R. felis* variants, and that it may well have resulted in symptoms typical of typhus-like flea-borne rickettsioses, including fever, fatigue, headache, maculopapular rash, sub-acute meningitis and pneumonia (SI 7)’.

Line 321-330, Materials and Methods: The first section could be divided into two. One for permits/provenience and one for DNA extraction to make it easier to read.

These sections have been separated, as requested.

Line 346: clarity needed regarding “...each individual (or accession number)”, did you study more than one individual/skeleton? What type of accession number is this referring to? Museum accession?

We have clarified the above accession number and sample-related issue as follows: ‘The skeletal remains of Ballito Bay A (‘BBayA’) belong to a juvenile individual excavated during the 1960s. The remains were curated at the Durban Museum, and then transferred to the KwaZulu-Natal Museum where these are now curated (accession No. 2009/007)². Permission for the sampling of the remains was obtained from the Council of the KwaZulu-Natal Museum. A sampling permit (No. 0014/06) was issued to M. Lombard under the KwaZulu-Natal Heritage Act No. 4 of 2008 and Section 38 (1) of the National Heritage Resources Act No. 25 of 1999. From the accessioned skeletal remains, analysed samples were extracted from the left petrous bone, right petrous bone and the upper left premolar (Table 1)’.

Line 406: please specify which genomes were used when referring to “multiple concatenated genomes”.

We have clarified this issue, which originates from an editing error, by replacing ‘multiple concatenated genomes, with ‘multiple concatenated genes’ in the manuscript text.

Reviewers' comments:

Reviewer #3 (Remarks to the Author):

Major revisions:

- Both Reviewer 1 and I referred to the use of MAPQ 37 in the competitive mapping as exclusionary of reads mapping equally well to conserved regions in several genomes. If you use mapping quality of 37 when you map to several genomes/references concatenated into one fasta file (i.e. competitive mapping) this means, as you state, that reads mapping equally well to more than one reference will be excluded from the mapping. This means that reads mapping to conserved regions shared by more than one genome (ie. core genes) will be excluded. To avoid this, you should have used MAPQ 0 (zero) instead in the competitive mapping. Given that you did seemingly manage to retrieve the core genes, I do not think you need to repeat these analyses. However, the limitations of your current approach need to be explicitly acknowledged in the main manuscript. You might test to see if using MAPQ 0 gives you a different result, as this would be the correct way to do it. You may have just been lucky that your reads were sufficiently divergent enough or had enough damage that they mapped only to one genome amongst the 126 that you used for the competitive mapping. It could also be related to the fact that you used BWA-mem which uses soft-clipping of the reads of reads when mapping (see issue with bwa-mem below). The use of MAPQ 37 is a point that others familiar with competitive mapping will pick up on.

- BWA-mem is used throughout the manuscript. Bwa-mem is meant for mapping long reads and uses local alignment which employs soft clipping of ends of reads.

From BWA man-page:

"The BWA-MEM algorithm performs local alignment. It may produce multiple primary alignments for different part of a query sequence."

"By default, BWA-MEM uses soft clipping for the primary alignment and hard clipping for supplementary alignments." (<https://www.mankier.com/1/bwa>)

Bwa-mem will therefore force contaminant reads to map. BWA-aln uses global alignment and does not use soft-clipping, which is why it is widely preferred for mapping ancient DNA. And is found to outperform bwa-mem. see Oliva et al. 2021 (<https://doi.org/10.1093/bib/bbab076>) and Oliva et al. 2021(<https://doi.org/10.1002/ece3.8297>). This problem is especially important to consider when working with ancient microbial DNA from metagenomic datasets such as yours which contains modern microbial DNA from the soil/burial environment that can easily cross-map. Even though there may not be closely related species in your shotgun data (ie. other Rickettsia species) there are often common genes that share a high degree of similarity between otherwise genetically distant bacterial species.

At minimum you need to provide the parameters you used for mapping with bwa-mem at every instance in the methods where you state that you used it.

- Quality of SNP calls: Do you exclude multiallelic SNPs? What is your cut-off for calling homozygous SNPs (ie. what percentage of reads covering a SNP must support either the reference or the variant call for it to be considered homozygous?) Did you do any type of analysis to quality control for the SNPs/alignment you use in your phylogenetic analyses? You state that you used snippy, what parameters did you use here? If the authors did not impose any quality control of the SNPs called with snippy, then I suggest that they investigate whether they have a high proportion of heterozygous/multiallelic sites in their BBayA genome. Additionally, the authors could for example run ClonalFrameML to clean up the MAFFT alignment to exclude recombinant sites. Many species of bacteria share conserved regions/genes that will easily map between species. This is especially important since you have used BWA-mem, which means you may have a higher chance of mapping contaminant reads due to the soft-clipping issue (see above).

- In line 136 they state that they were able to “assemble 99.53% of the ancient *R. felis* genome”. This is referring to the completeness statistic provided by CheckM, which is based on the detection of single core genes (SCGs). It does not refer to the true completeness/contamination of the whole genome. I suggest the authors read the explanation of these stats provided here: <https://microbe.net/2017/12/13/why-genome-completeness-and-contamination-estimates-are-more-complicated-than-you-think/>, as I do not think that they fully grasp what these statistics represent. An adjustment in phrasing at line 136 and any other relevant place in the manuscript is recommended.
- By recruiting reads through mapping to the 126 *Rickettsia* genomes, your de-novo assembly is not necessarily going to be a true representation of the ancient *R. felis* genome. If your 2000-year-old genome contained genomic regions that are not present in the modern diversity of *R. felis/Rickettsia*, then you will have missed these regions. This limitation needs to be acknowledged in the manuscript.
- Why are different coverages for the *R. felis* genome reported? In line 89 (and elsewhere) a genome coverage of 28.97-fold is reported, while in line 135 11.41-fold coverage for the chromosome is reported. For the 28.97-fold, does this include the plasmids? While 11.41-fold is only for the chromosome? If the plasmids are included in the 28.97-fold, then this should be explicitly specified. It is not customary to include them when talking about average fold coverage for bacteria, since the plasmids are often present at very high copy numbers compared to the chromosome and will give a false sense of inflated coverage.
- As another quality check of your assembled contigs, I suggest you do apply pyDamage (<https://github.com/maxibor/pydamage>) to assess the aDNA damage levels of your assembled contigs to double check that they are indeed ancient.
- Did the authors employ a length restriction of the contigs you included in your analysis? Many researchers use and assume an automatic cut-off of 1000 bp, where all contigs below 1000 bp are discarded. Therefore, many programs calculating stats for contigs (e.g. N50) report values calculated based off of only contigs above 1000 bp. Quast does this, and I took a quick look at CoverM, which you used, and it looks like it might do it too. If this is the case and you used contigs below 1000 bp in length, then you should specify in the manuscript what you did, or did not, exclude in the contigs you classify as your genome assembly.
- Line 132-134: The authors state: “This genomic structure allows high mapping specificity across the *R. felis* genome, which allowed us to infer the presence and absence of genomic regions from the level of coverage observed after mapping the raw datasets to the reference genomes”. If you used MAPQ 37 for this mapping you cannot properly assess for absence/presence, you need to use MAPQ 0.
- Date for RefSeq download used to build the Kraken2 database needs to be supplied in the main manuscript, the authors supplied it in response to my previous request in their rebuttal. It should be added in line 80-81. The RefSeq database is continually updated and it is common practice to supply the download date in the manuscript.
- Line 406-407: is “concatenated genes” a typo and supposed to be “concatenated genomes” here?
- Line 425-426: I do not fully understand what is meant by this sentence: “b) the percentage of reads used for the assembly were > 89% for the *R. felis* genomes, while numbers < 72% were found for the other species”, If this was a competitive mapping where the data was mapped to a multifasta containing all genomes, then wouldn't the percentage of reads amount to 100%? Can the authors make it clearer what is meant in the manuscript.
- Table S3: I think something has gone wrong with this table. The first column shown is collection date. There are no GCF numbers, accession codes or strain names supplied in the version of the table

that I have access to. This table is not interpretable without this information. Furthermore, the "collection source" column is unclear. Is "reference" supposed to indicate RefSeq? If there is a dash (-) in this column where did you get the genome from?

Minor revisions:

- Line 20: please specify that you are talking about human "genomic evidence" here.
- Table 1 (page 4), what statistic is used for genomic coverage (x)? mean? Median? Dis the reference you mapped to include plasmids? If so, this needs to be stated.
- Line 127: You mention doing analysis to detect plasmid rearrangements, but you never mention it again. Did this analysis yield any results? If it was not done, it should be excluded.
- Line 128: specify whether you mapped the data individually to each reference for the 126 genomes or whether you did a competitive mapping (ie. where you concatenate all references into one multifasta and mapped to that)
- Line 156: are you referring to de-novo reconstruction here or mapping? This should be specified throughout the manuscript. Sometimes people use reconstruction to refer to mapping. And since you have done both, you should be clear when you are talking about what.
- Figure 3 (page 7): the branches in sub-figure C do not show up well in the PDF.
- Line 331-332: DNA Away does not contain bleach. It contains Sodium Hydroxide which is caustic soda/lye. Therefore, the sentence: "Further handling of the specimens was done in a bleach decontaminated (DNA Away, ThermoScientific) enclosed sampling tent..." does not make sense.
- DNA Away is referred to interchangeably as "DNA Away" (correct) and "DNA-away" throughout the methods. And "ThermoScientific" is referred to in several different ways throughout the methods. These names should be consistent throughout.
- Line 375: A reference (Schlebusch et al. 2017) is referred to for methods. This is not an open access paper. If methods listed in Schlebusch et al. 2017, beyond mapdamage which is described, were used in this manuscript, then those methods should be further explained in this manuscript.
- Line 432-434: Repetition of the same information. Also, you need to state that the completeness and contamination stats are derived from CheckM, and not coverm which is the tool most recently referred to prior to these sentences.

9 December 2022

Reviewer Reply Letter

Re: *Rickettsia felis* DNA recovered from a child who lived in southern Africa 2,000 years ago.

Dear Reviewer,

We thank you for your time in providing us with such a detailed and constructive review concerning the above manuscript, following submission to Nature Communications Biology.

Following extensive revision of the manuscript, including various re-analyses of the datasets, please see our reply to the commentary provided, below.

- 1. Both Reviewer 1 and I referred to the use of MAPQ 37 in the competitive mapping as exclusionary of reads mapping equally well to conserved regions in several genomes. If you use mapping quality of 37 when you map to several genomes/references concatenated into one fasta file (i.e. competitive mapping) this means, as you state, that reads mapping equally well to more than one reference will be excluded from the mapping. This means that reads mapping to conserved regions shared by more than one genome (i.e. core genes) will be excluded. To avoid this, you should have used MAPQ 0 (zero) instead in the competitive mapping. Given that you did seemingly manage to retrieve the core genes, I do not think you need to repeat these analyses. However, the limitations of your current approach need to be explicitly acknowledged in the main manuscript. You might test to see if using MAPQ 0 gives you a different result, as this would be the correct way to do it. You may have just been lucky that your reads were sufficiently divergent enough or had enough damage that they mapped only to one genome amongst the 126 that you used for the competitive mapping. It could also be related to the fact that you used BWA-mem which uses soft-clipping of the reads of reads when mapping (see issue with bwa-mem below). The use of MAPQ 37 is a point that others familiar with competitive mapping will pick up on. BWA-mem is used throughout the manuscript. Bwa-mem is meant for mapping long reads and uses local alignment which employs soft clipping of ends of reads.**

We thank the reviewer and editor for this comment. Here we must clarify some points. There are several mapping steps and tools that were used throughout the manuscript, and these are now clarified in the respective sections in L78-87, L396-408 and L424-455. To perform the competitive mapping against the pathogen custom database, we used the Bowtie2 software (-very-sensitive) which doesn't discard reads equally mapping to a genome. The randomness of the seed procedure

of this pre-set of Bowtie2 helps to find the best scores to the alignments without filtering. This methodology is now indicated in the Materials and Methods section in the revised manuscript.

The competitive mapping was performed just in a first step to identify the most likely pathogens to be present in the samples, and not to extract the reads and perform the assembly. For that purpose, a mapping with `bwa aln -n 0.02 -l 1024` was used to identify the *R. felis* reads against two references and then to extract those reads (in a non-competitive mapping) to further assemble them (L424-429). The later comparison of the reads used for the assembly and their mapping against the 126 other *Rickettsia* genomes was performed in competitive as well non-competitive ways but just to report the numbers of mapped reads. Therefore, to summarise, the reads from the un-repaired and repaired libraries were aligned to the *R. felis* LSU-Lb and URRWXcal2 strains as reference genomes during the BWA v0.6.2-r126 alignment (`bwa aln -n 0.02 -l 1024`) to recover the reads and using them for the assembly, therefore any reads that mapped equally to both references were not discarded.

- 2. From BWA man-page: The BWA-MEM algorithm performs local alignment. It may produce multiple primary alignments for different part of a query sequence. By default, BWA-MEM uses soft clipping for the primary alignment and hard clipping for supplementary alignments. (<https://www.mankier.com/1/bwa>). Bwa-mem will therefore force contaminant reads to map. BWA-aln uses global alignment and does not use soft-clipping, which is why it is widely preferred for mapping ancient DNA. And is found to outperform bwa-mem. see Oliva et al. 2021 (<https://doi.org/10.1093/bib/bbab076>) and Oliva et al. 2021 (<https://doi.org/10.1002/ece3.8297>). This problem is especially important to consider when working with ancient microbial DNA from metagenomic datasets such as yours which contains modern microbial DNA from the soil/burial environment that can easily cross-map. Even though there may not be closely related species in your shotgun data (i.e. other *Rickettsia* species) there are often common genes that share a high degree of similarity between otherwise genetically distant bacterial species. At minimum you need to provide the parameters you used for mapping with bwa-mem at every instance in the methods where you state that you used it.**

The authors acknowledge this comment, which required a more comprehensive explanation of our methods, as the following mapping steps indicate:

1. In our previous version we stated that bwa-mem was used to map the *H. sapiens* reads, this is still the method to filter the human reads.
2. After that, a competitive mapping was performed to identify the most likely pathogens from the custom NCBI downloaded reference genomes including just 2 *R. felis* genomes (identified with the Kraken analyses). That competitive mapping was performed with Bowtie2 software.
3. To extract the reads, our previous draft said that bwa software was used, however the parameters were not given and now are explicit in the current version (`bwa aln -n 0.02 -l 1024`) which is accordingly with what reviewers suggested.
4. In the further analyses against the 126 *Rickettsia* genomes, the software bwa-mem was used again as it was just for coverage calculation purposes with the CoverM software.

Again, we acknowledge the commentaries as were useful to be more precise in the parameters used in the different software and to clarify the order of the procedures. We believe all this is now clearly stated in the revised manuscript.

- 3. Quality of SNP calls: Do you exclude multiallelic SNPs? What is your cut-off for calling homozygous SNPs (i.e. what percentage of reads covering a SNP must support either the reference or the variant call for it to be considered homozygous?) Did you do any type of analysis to quality control for the SNPs/alignment you use in your phylogenetic analyses? You state that you used snippy, what parameters did you use here? If the authors did not impose any quality control of the SNPs called with snippy, then I suggest that they investigate whether they have a high proportion of heterozygous/multiallelic sites in their BBayA genome. Additionally, the authors could for example run ClonalFrameML to clean up the MAFFT alignment to exclude recombinant sites. Many species of bacteria share conserved regions/genes that will easily map between species. This is especially important since you have used BWA-mem, which means you may have a higher chance of mapping contaminant reads due to the soft-clipping issue (see above).**

We thank the Reviewer for this valuable comment. We tested ClonalFrameML (Didelot and Wilson, 2015) with the option ‘-rescale_no_recombination true’ against the codon alignment, i.e., the base of the phylogenomic trees. We obtained a similar topology for the entire tree with corrected branch lengths. This same topology was also recovered for the *R. felis* clade that we previously had in our first submitted manuscript. We furthermore performed an analysis using the kSNP V3.1 to identify and construct the phylogenetic tree based on the SNPs in the selected *Rickettsia* genomes (<https://academic.oup.com/bioinformatics/article/31/17/2877/183216>). There were 186 SNPs in the new analysis. First, the optimum Kmer value was identified for all the selected *Rickettsia* genomes using the Kchooser. 31 mer was identified as the best Kmer value for the kSNP analysis. Then kSNP analysis was performed using the ‘kSNP3 -in fasta-input-list -ML -CPU 4 -outdir kSNP-out -k 31 -annotate annotate-list’. This information is given now in L429-434. In the Snippy analysis, no multiallelic SNPs were identified. We therefore performed the SNP analysis using kSNP v3.1.

- 4. In line 136 they state that they were able to “assemble 99.53% of the ancient *R. felis* genome”. This is referring to the completeness statistic provided by CheckM, which is based on the detection of single core genes (SCGs). It does not refer to the true completeness/contamination of the whole genome. I suggest the authors read the explanation of these stats provided here: <https://microbe.net/2017/12/13/why-genome-completeness-and-coamination-estimates-are-more-complicated-than-you-think/>, as I do not think that they fully grasp what these statistics represent. An adjustment in phrasing at line 136 and any other relevant place in the manuscript is recommended.**

The authors agree with the Reviewer and the phrases were corrected to state that ‘We were able to recover reads homologous to 94.73% of the *R. felis* LSU-Lb genome (bwa-mem --min-read-percent-identity 95 --min-read-aligned-percent 50), while the ancient *R. felis* BBayA genome assembly is 99.53% complete according to checkM software’ in L135-138, L238-239, and L441-442.

- 5. By recruiting reads through mapping to the 126 Rickettsia genomes, your de-novo assembly is not necessarily going to be a true representation of the ancient R. felis genome. If your 2000-year-old genome contained genomic regions that are not present in the modern diversity of R. felis/Rickettsia, then you will have missed these regions. This limitation needs to be acknowledged in the manuscript.**

We agree with the Reviewer that this limitation needs to be acknowledged in the manuscript, and we have done so in L446-448. Although we could perform additional analyses to address this issue, the best option would be to re-assemble the ancient bacterial reads and then to bin or map the contigs to the modern genomes. However, this would imply to perform all the analyses from 'scratch' and given our current funding and project requirements, this is not a viable option in terms of this manuscript.

- 6. Why are different coverages for the R. felis genome reported? In line 89 (and elsewhere) a genome coverage of 28.97-fold is reported, while in line 135 11.41-fold coverage for the chromosome is reported. For the 28.97-fold, does this include the plasmids? While 11.41-fold is only for the chromosome? If the plasmids are included in the 28.97-fold, then this should be explicitly specified. It is not customary to include them when talking about average fold coverage for bacteria, since the plasmids are often present at very high copy numbers compared to the chromosome and will give a false sense of inflated coverage.**

We acknowledge this comment. The 28.97-fold coverage indicated in Table 1, is a simple addition to the coverage of the LSU-Lb genome obtained from each sample, while the 11.41-fold coverage was an average of all samples without the plasmids considered. In the current version, we calculated the coverage of the BBayA genome over the mapped reads used for the assembly being 55.1X and shown in L139. These results were obtained using the CoverM software with the options '-p bwa-mem -m trimmed_mean --min-read-percent-identity 95 --min-read-aligned-percent 50'. This method is the trimmed value, removing the 5% from both extremes (low and high coverage) across the genome. The mean coverage without the trimming values was 61.7074, including the plasmid. We also calculated the mean/trimmed-mean coverage values for the mapping using the same parameters as above for the plasmids pRF and pLbAr from the LSU-Lb genome (as the plasmid pRF in our assembly is fragmented). The coverage of the plasmid pRF are 132.25534/106.19299, while the coverages for the plasmid pLbAr are 0/0 and are given in L54-155.

- 7. As another quality check of your assembled contigs, I suggest you do apply PyDamage (<https://github.com/maxibor/pydamage>) to assess the aDNA damage levels of your assembled contigs to double check that they are indeed ancient.**

We applied PyDamage to the BBayA assembly and the LSU-Lb genome, and the obtained results indicate that at least 36.9 % and 60.75 %, respectively, of the assemblies corresponded to aDNA according to the strict q-values (<0.05) with an accuracy >0.5 of the test. As at least a third of the assembled genome comes from aDNA, we strongly believe that this assembly is most likely to be ancient than a recent contamination of the sample. This information is now given in L398-402. The sequencing library was prepared in two parts, namely 1) no terminal damage repair and 2) damage-repaired. We highlight here that the DNA damage was performed using the reads with 'no terminal damage repair'. The extraction of reads and assembly was performed with the reads which were processed for the damage repair protocol during the library preparation and sequencing.

- 8. Did the authors employ a length restriction of the contigs you included in your analysis? Many researchers use and assume an automatic cut-off of 1000 bp, where all contigs below 1000 bp are discarded. Therefore, many programs calculating stats for contigs (e.g. N50) report values calculated based off of only contigs above 1000 bp. Quast does this, and I took a quick look at CoverM, which you used, and it looks like it might do it too. If this is the case and you used contigs below 1000 bp in length, then you should specify in the manuscript what you did, or did not, exclude in the contigs you classify as your genome assembly.**

We did not employ a length restriction in the previous version. In our assembly, a total of 36 contigs are < 1000 bp, totalling 111 contigs. Some contigs do not map to the LSU-Lb assembly, and they are < 3000 bp. There are also additional contigs < 1000 bp that map to LSU-Lb, but they don't have so much quality for study in the scientific community. Besides, the LSU-Lb assembly is composed of 43 contigs and 13 of them are < 1000 bp, suggesting that some of the *R. felis* genomes in the database are not of the highest quality. Returning to the small contigs, these have been removed accordingly and the analyses presented throughout the revision are without these small and doubtful contigs. The new genome stats are given in L457-460. Regarding the concerns of the Reviewer, CoverM can use contigs below < 1000 bp.

- 9. Line 132-134: The authors state: “This genomic structure allows high mapping specificity across the *R. felis* genome, which allowed us to infer the presence and absence of genomic regions from the level of coverage observed after mapping the raw datasets to the reference genomes”. If you used MAPQ 37 for this mapping you cannot properly assess for absence/presence, you need to use MAPQ 0.**

We have removed this sentence and we have clarified our mapping steps as indicated in our responses above.

- 10. Date for RefSeq download used to build the Kraken2 database needs to be supplied in the main manuscript the authors supplied it in response to my previous request in their rebuttal. It should be added in line 80-81. The RefSeq database is continually updated and it is common practice to supply the download date in the manuscript.**

As indicated in the revised manuscript text, the Kraken2 database was downloaded on 30 October 2019 and NCBI-RefSeq database was downloaded on 14 March 2018. This information is given in L74-76.

- 11. Line 406-407: is “concatenated genes” a typo and supposed to be “concatenated genomes” here?**

We have modified our text for clarity. In L182 we now state that ‘Multiple sequence alignment of the codons of 138 concatenated core genes for the 127 *Rickettsia* genomes was used for the phylogenomic reconstruction using the maximum likelihood method (ML)’.

- 12. Line 425-426: I do not fully understand what is meant by this sentence: “(b) the percentage of reads used for the assembly were > 89% for the *R. felis* genomes, while numbers < 72% were found for the other species”, If this was a competitive mapping where the data was**

mapped to a multifasta containing all genomes, then wouldn't the percentage of reads amount to 100%? Can the authors make it clearer what is meant in the manuscript?

We agree with the Reviewer and have corrected this sentence in the manuscript to state that 'b) in a non-competitive mapping (bwa-mem --min-read-percent-identity 95 --min-read-aligned-percent 80), the percentage of reads used in the BBayA assembly were at least > 89% for any of the *R. felis* genomes, while numbers < 72% were found for the other species...'. This is given in L451-454.

13. Table S3: I think something has gone wrong with this table. The first column shown is collection date. There are no GCF numbers, accession codes or strain names supplied in the version of the table that I have access to. This table is not interpretable without this information. Furthermore, the "collection source" column is unclear. Is "reference" supposed to indicate RefSeq? If there is a dash (-) in this column where did you get the genome from?

We thank the Reviewer for noting this. All the required information was in the previous version of the table, but it was displayed after column P. The strain name was given in column A, while the NCBI RefSeq accession number was in column C. For clarity, we searched for the origin and date of the samples in different databases to calculate the temporal signal, and therefore we obtained the collection date of the strains from different sources: NCBI, LPSN or the publication of the original samples (DOI's provided). Regarding the title of the column "collection source", it was changed by 'collection_date_source' as we obtained the collection date from different sources and in some cases, we had to choose the most confident.

14. Line 20: please specify that you are talking about human "genomic evidence" here.

This has been corrected.

15. Table 1 (page 4), what statistic is used for genomic coverage (x)? mean? Median? Did the reference you mapped to include plasmids? If so, this needs to be stated.

We corrected this in Table 1.

16. Line 127: You mention doing analysis to detect plasmid rearrangements, but you never mention it again. Did this analysis yield any results? If it was not done, it should be excluded.

We performed several analyses regarding the distribution of plasmid across the 5 *R. felis* genomes. Some of the results are given in L147-148 and L195-203. We found that the main differences between genomes was in the presence/absence of plasmids, seeming the plasmid pRF ancestral (and distributed in 4 genomes) while in the URRWXCal2 strain there was a duplication of part of this pRF plasmid and in LSU-Lb (closer to BBayA genome) an acquisition of the pLBaR plasmid through HGT. Note that the host of the LSU-Lb strain is not the cat flea (*C. felis*), but the booklouse or *Liposcelis bostrychophilus*, suggesting that this BBayA genome could have been acquired by another host besides fleas.

17. Line 128: specify whether you mapped the data individually to each reference for the 126 genomes or whether you did a competitive mapping (i.e. where you concatenate all references into one multifasta and mapped to that).

This sentence was changed regarding the comments from above. Regarding the comparison with the 126 *Rickettsia* genomes, we performed competitive as well non-competitive mapping and is clarified now in Mat/Met as said in comments above.

18. Line 156: are you referring to de-novo reconstruction here or mapping? This should be specified throughout the manuscript. Sometimes people use reconstruction to refer to mapping. And since you have done both, you should be clear when you are talking about what.

We referred to the assembly, and this is now indicated in L161.

19. Figure 3 (page 7): the branches in sub-figure C do not show up well in the PDF.

We have improved sub-figure C accordingly.

20. Line 331-332: DNA Away does not contain bleach. It contains Sodium Hydroxide which is caustic soda/lye. Therefore, the sentence: “Further handling of the specimens was done in a bleach decontaminated (DNA Away, ThermoScientific) enclosed sampling tent...” does not make sense.

This has been corrected to state that ‘Further handling of the specimens was done in a bleach-decontaminated, also using DNA-Away (Thermo Scientific) enclosed sampling tent with adherent gloves (Captair Pyramide portable isolation enclosure, Erlab)’.

21. DNA Away is referred to interchangeably as “DNA Away” (correct) and “DNA-away” throughout the methods. And “ThermoScientific” is referred to in several different ways throughout the methods. These names should be consistent throughout.

This has been corrected throughout the manuscript text.

22. Line 375: A reference (Schlebusch et al. 2017) is referred to for methods. This is not an open access paper. If methods listed in Schlebusch et al. 2017, beyond mapDamage which is described, were used in this manuscript, then those methods should be further explained in this manuscript.

We indicate that ‘The manuscript from which the data analysed in this study is derived, i.e., Schlebusch et al. 2017, is available at https://www.researchgate.net/publication/320101464_Southern_African_ancient_genomes_estimate_modern_human_divergence_to_350000-260000_years_ago’. This is now shown in L335-338.

23. Line 432-434: Repetition of the same information. Also, you need to state that the completeness and contamination stats are derived from CheckM, and not coverm which is the tool most recently referred to prior to these sentences.

We have re-structured this section to clarify the analyses performed using CheckM and CoverM, indicating that ‘The quality of all the *Rickettsia* genomes was evaluated with the CheckM v1.1.3 software package⁷³ (Table S5). Of the available NCBI RefSeq genomes used, some displayed lower completeness values (e.g., 90.97 % for GCF_000964995.1) and higher contamination values (e.g., 7.04 % for GCF_000696365.1). The completeness was 99.53%, and contamination 0.47% according to CheckM software. From the other *R. felis* strain genomes, only those from the strains

Pedreira and URRWXCal2 achieved 100% completeness, with 0% and 0.47% contamination, respectively. The remaining two *R. felis* strains, namely LSU-Lb and LSU, presented lower completeness and contamination values (at 97.47% and 1.42%, and 94.14% and 0.71%, respectively (Table S3)). Also, we indicate that 'Following this, the reads that were used to assemble the *R. felis* genome were mapped against the obtained assembly, as well the other 126 *Rickettsia* genomes, using Coverm v0.6.1 (<https://github.com/wwood/CoverM>) with the BWA-MEM mapping tool, resulting in a) a competitive mapping with the five *R. felis* genomes recruiting over 80% of reads, b) in a non-competitive mapping (bwa-mem --min-read-percent-identity 95 --min-read-aligned-percent 80), the percentage of reads used in the BBayA assembly were at least > 89% for any of the *R. felis* genomes, while numbers < 72% were found for the other species, and c) the percentage of bases of the genome covered for at least 1 read at 95% identity and 80% of read alignment, was > 98% for the *R. felis* genomes, excepting the *R. felis* str. Pedreira with 94%, GCF_000964665.1, while for other species it was < 80% (e.g., GCF_000828125.2, *R. asembonensis*). The obtained *R. felis* BBayA genome consists of 1,512,774 bases and 69 contigs, with a N50 of 42,410 bases and the longest contig comprising 121,989 bases. It exhibits a GC value of 32.5% and a coding density of 84.58% for 1,561 predicted proteins'.

5 January 2023

Reviewer Reply Letter

Re: *Rickettsia felis* DNA recovered from a child who lived in southern Africa 2,000 years ago.

Dear Reviewer,

We thank you for your time in providing us with such a detailed and constructive review concerning the above manuscript, following submission to Nature Communications Biology.

Following extensive revision of the manuscript, including various re-analyses of the datasets, please see our reply to the commentary provided, below.

- 1. Both Reviewer 1 and I referred to the use of MAPQ 37 in the competitive mapping as exclusionary of reads mapping equally well to conserved regions in several genomes. If you use mapping quality of 37 when you map to several genomes/references concatenated into one fasta file (i.e. competitive mapping) this means, as you state, that reads mapping equally well to more than one reference will be excluded from the mapping. This means that reads mapping to conserved regions shared by more than one genome (i.e. core genes) will be excluded. To avoid this, you should have used MAPQ 0 (zero) instead in the competitive mapping. Given that you did seemingly manage to retrieve the core genes, I do not think you need to repeat these analyses. However, the limitations of your current approach need to be explicitly acknowledged in the main manuscript. You might test to see if using MAPQ 0 gives you a different result, as this would be the correct way to do it. You may have just been lucky that your reads were sufficiently divergent enough or had enough damage that they mapped only to one genome amongst the 126 that you used for the competitive mapping. It could also be related to the fact that you used BWA-mem which uses soft-clipping of the reads of reads when mapping (see issue with bwa-mem below). The use of MAPQ 37 is a point that others familiar with competitive mapping will pick up on. BWA-mem is used throughout the manuscript. Bwa-mem is meant for mapping long reads and uses local alignment which employs soft clipping of ends of reads.**

We thank the reviewer and editor for this comment. Here we must clarify some points. There are several mapping steps and tools that were used throughout the manuscript, and these are now clarified in the respective sections in L78-87, L396-408 and L424-455. To perform the competitive mapping against the pathogen custom database, we used the Bowtie2 software (-very-sensitive) which doesn't discard reads equally mapping to a genome. The randomness of the seed procedure

of this pre-set of Bowtie2 helps to find the best scores to the alignments without filtering. This methodology is now indicated in the Materials and Methods section in the revised manuscript.

The competitive mapping was performed just in a first step to identify the most likely pathogens to be present in the samples, and not to extract the reads and perform the assembly. For that purpose, a mapping with `bwa aln -n 0.02 -l 1024` was used to identify the *R. felis* reads against two references and then to extract those reads (in a non-competitive mapping) to further assemble them (L424-429). The later comparison of the reads used for the assembly and their mapping against the 126 other *Rickettsia* genomes was performed in competitive as well non-competitive ways but just to report the numbers of mapped reads. Therefore, to summarise, the reads from the un-repaired and repaired libraries were aligned to the *R. felis* LSU-Lb and URRWXcal2 strains as reference genomes during the BWA v0.6.2-r126 alignment (`bwa aln -n 0.02 -l 1024`) to recover the reads and using them for the assembly, therefore any reads that mapped equally to both references were not discarded.

- 2. From BWA man-page: The BWA-MEM algorithm performs local alignment. It may produce multiple primary alignments for different part of a query sequence. By default, BWA-MEM uses soft clipping for the primary alignment and hard clipping for supplementary alignments. (<https://www.mankier.com/1/bwa>). Bwa-mem will therefore force contaminant reads to map. BWA-aln uses global alignment and does not use soft-clipping, which is why it is widely preferred for mapping ancient DNA. And is found to outperform bwa-mem. see Oliva et al. 2021 (<https://doi.org/10.1093/bib/bbab076>) and Oliva et al. 2021 (<https://doi.org/10.1002/ece3.8297>). This problem is especially important to consider when working with ancient microbial DNA from metagenomic datasets such as yours which contains modern microbial DNA from the soil/burial environment that can easily cross-map. Even though there may not be closely related species in your shotgun data (i.e. other *Rickettsia* species) there are often common genes that share a high degree of similarity between otherwise genetically distant bacterial species. At minimum you need to provide the parameters you used for mapping with bwa-mem at every instance in the methods where you state that you used it.**

The authors acknowledge this comment, which required a more comprehensive explanation of our methods, as the following mapping steps indicate:

1. In our previous version we stated that bwa-mem was used to map the *H. sapiens* reads, this is still the method to filter the human reads.
2. After that, a competitive mapping was performed to identify the most likely pathogens from the custom NCBI downloaded reference genomes including just 2 *R. felis* genomes (identified with the Kraken analyses). That competitive mapping was performed with Bowtie2 software.
3. To extract the reads, our previous draft said that bwa software was used, however the parameters were not given and now are explicit in the current version (`bwa aln -n 0.02 -l 1024`) which is accordingly with what reviewers suggested.
4. In the further analyses against the 126 *Rickettsia* genomes, the software bwa-mem was used again as it was just for coverage calculation purposes with the CoverM software.

Again, we acknowledge the commentaries as were useful to be more precise in the parameters used in the different software and to clarify the order of the procedures. We believe all this is now clearly stated in the revised manuscript.

- 3. Quality of SNP calls: Do you exclude multiallelic SNPs? What is your cut-off for calling homozygous SNPs (i.e. what percentage of reads covering a SNP must support either the reference or the variant call for it to be considered homozygous?) Did you do any type of analysis to quality control for the SNPs/alignment you use in your phylogenetic analyses? You state that you used snippy, what parameters did you use here? If the authors did not impose any quality control of the SNPs called with snippy, then I suggest that they investigate whether they have a high proportion of heterozygous/multiallelic sites in their BBayA genome. Additionally, the authors could for example run ClonalFrameML to clean up the MAFFT alignment to exclude recombinant sites. Many species of bacteria share conserved regions/genes that will easily map between species. This is especially important since you have used BWA-mem, which means you may have a higher chance of mapping contaminant reads due to the soft-clipping issue (see above).**

We thank the Reviewer for this valuable comment. We tested ClonalFrameML (Didelot and Wilson, 2015) with the option ‘-rescale_no_recombination true’ against the codon alignment, i.e., the base of the phylogenomic trees. We obtained a similar topology for the entire tree with corrected branch lengths. This same topology was also recovered for the *R. felis* clade that we previously had in our first submitted manuscript. We furthermore performed an analysis using the kSNP V3.1 to identify and construct the phylogenetic tree based on the SNPs in the selected *Rickettsia* genomes (<https://academic.oup.com/bioinformatics/article/31/17/2877/183216>). There were 186 SNPs in the new analysis. First, the optimum Kmer value was identified for all the selected *Rickettsia* genomes using the Kchooser. 31 mer was identified as the best Kmer value for the kSNP analysis. Then kSNP analysis was performed using the ‘kSNP3 -in fasta-input-list -ML -CPU 4 -outdir kSNP-out -k 31 -annotate annotate-list’. This information is given now in L429-434. In the Snippy analysis, no multiallelic SNPs were identified. We therefore performed the SNP analysis using kSNP v3.1.

- 4. In line 136 they state that they were able to “assemble 99.53% of the ancient *R. felis* genome”. This is referring to the completeness statistic provided by CheckM, which is based on the detection of single core genes (SCGs). It does not refer to the true completeness/contamination of the whole genome. I suggest the authors read the explanation of these stats provided here: <https://microbe.net/2017/12/13/why-genome-completeness-and-coamination-estimates-are-more-complicated-than-you-think/>, as I do not think that they fully grasp what these statistics represent. An adjustment in phrasing at line 136 and any other relevant place in the manuscript is recommended.**

The authors agree with the Reviewer and the phrases were corrected to state that ‘We were able to recover reads homologous to 94.73% of the *R. felis* LSU-Lb genome (bwa-mem --min-read-percent-identity 95 --min-read-aligned-percent 50), while the ancient *R. felis* BBayA genome assembly is 99.53% complete according to checkM software’ in L135-138, L238-239, and L441-442.

- 5. By recruiting reads through mapping to the 126 Rickettsia genomes, your de-novo assembly is not necessarily going to be a true representation of the ancient R. felis genome. If your 2000-year-old genome contained genomic regions that are not present in the modern diversity of R. felis/Rickettsia, then you will have missed these regions. This limitation needs to be acknowledged in the manuscript.**

We agree with the Reviewer that this limitation needs to be acknowledged in the manuscript, and we have done so in L446-448. Although we could perform additional analyses to address this issue, the best option would be to re-assemble the ancient bacterial reads and then to bin or map the contigs to the modern genomes. However, this would imply to perform all the analyses from 'scratch' and given our current funding and project requirements, this is not a viable option in terms of this manuscript.

- 6. Why are different coverages for the R. felis genome reported? In line 89 (and elsewhere) a genome coverage of 28.97-fold is reported, while in line 135 11.41-fold coverage for the chromosome is reported. For the 28.97-fold, does this include the plasmids? While 11.41-fold is only for the chromosome? If the plasmids are included in the 28.97-fold, then this should be explicitly specified. It is not customary to include them when talking about average fold coverage for bacteria, since the plasmids are often present at very high copy numbers compared to the chromosome and will give a false sense of inflated coverage.**

We acknowledge this comment. The 28.97-fold coverage indicated in Table 1, is a simple addition to the coverage of the LSU-Lb genome obtained from each sample, while the 11.41-fold coverage was an average of all samples without the plasmids considered. In the current version, we calculated the coverage of the BBayA genome over the mapped reads used for the assembly being 55.1X and shown in L139. These results were obtained using the CoverM software with the options '-p bwa-mem -m trimmed_mean --min-read-percent-identity 95 --min-read-aligned-percent 50'. This method is the trimmed value, removing the 5% from both extremes (low and high coverage) across the genome. The mean coverage without the trimming values was 61.7074, including the plasmid. We also calculated the mean/trimmed-mean coverage values for the mapping using the same parameters as above for the plasmids pRF and pLbAr from the LSU-Lb genome (as the plasmid pRF in our assembly is fragmented). The coverage of the plasmid pRF are 132.25534/106.19299, while the coverages for the plasmid pLbAr are 0/0 and are given in L54-155.

- 7. As another quality check of your assembled contigs, I suggest you do apply PyDamage (<https://github.com/maxibor/pydamage>) to assess the aDNA damage levels of your assembled contigs to double check that they are indeed ancient.**

We applied PyDamage to the BBayA assembly and the LSU-Lb genome, and the obtained results indicate that at least 36.9 % and 60.75 %, respectively, of the assemblies corresponded to aDNA according to the strict q-values (<0.05) with an accuracy >0.5 of the test. As at least a third of the assembled genome comes from aDNA, we strongly believe that this assembly is most likely to be ancient than a recent contamination of the sample. This information is now given in L398-402. The sequencing library was prepared in two parts, namely 1) no terminal damage repair and 2) damage-repaired. We highlight here that the DNA damage was performed using the reads with 'no terminal damage repair'. The extraction of reads and assembly was performed with the reads which were processed for the damage repair protocol during the library preparation and sequencing.

- 8. Did the authors employ a length restriction of the contigs you included in your analysis? Many researchers use and assume an automatic cut-off of 1000 bp, where all contigs below 1000 bp are discarded. Therefore, many programs calculating stats for contigs (e.g. N50) report values calculated based off of only contigs above 1000 bp. Quast does this, and I took a quick look at CoverM, which you used, and it looks like it might do it too. If this is the case and you used contigs below 1000 bp in length, then you should specify in the manuscript what you did, or did not, exclude in the contigs you classify as your genome assembly.**

We did not employ a length restriction in the previous version. In our assembly, a total of 36 contigs are < 1000 bp, totalling 111 contigs. Some contigs do not map to the LSU-Lb assembly, and they are < 3000 bp. There are also additional contigs < 1000 bp that map to LSU-Lb, but they don't have so much quality for study in the scientific community. Besides, the LSU-Lb assembly is composed of 43 contigs and 13 of them are < 1000 bp, suggesting that some of the *R. felis* genomes in the database are not of the highest quality. Returning to the small contigs, these have been removed accordingly and the analyses presented throughout the revision are without these small and doubtful contigs. The new genome stats are given in L457-460. Regarding the concerns of the Reviewer, CoverM can use contigs below < 1000 bp.

- 9. Line 132-134: The authors state: “This genomic structure allows high mapping specificity across the *R. felis* genome, which allowed us to infer the presence and absence of genomic regions from the level of coverage observed after mapping the raw datasets to the reference genomes”. If you used MAPQ 37 for this mapping you cannot properly assess for absence/presence, you need to use MAPQ 0.**

We have removed this sentence and we have clarified our mapping steps as indicated in our responses above.

- 10. Date for RefSeq download used to build the Kraken2 database needs to be supplied in the main manuscript the authors supplied it in response to my previous request in their rebuttal. It should be added in line 80-81. The RefSeq database is continually updated and it is common practice to supply the download date in the manuscript.**

As indicated in the revised manuscript text, the Kraken2 database was downloaded on 30 October 2019 and NCBI-RefSeq database was downloaded on 14 March 2018. This information is given in L74-76.

- 11. Line 406-407: is “concatenated genes” a typo and supposed to be “concatenated genomes” here?**

We have modified our text for clarity. In L182 we now state that ‘Multiple sequence alignment of the codons of 138 concatenated core genes for the 127 *Rickettsia* genomes was used for the phylogenomic reconstruction using the maximum likelihood method (ML)’.

- 12. Line 425-426: I do not fully understand what is meant by this sentence: “(b) the percentage of reads used for the assembly were > 89% for the *R. felis* genomes, while numbers < 72% were found for the other species”, If this was a competitive mapping where the data was**

mapped to a multifasta containing all genomes, then wouldn't the percentage of reads amount to 100%? Can the authors make it clearer what is meant in the manuscript?

We agree with the Reviewer and have corrected this sentence in the manuscript to state that 'b) in a non-competitive mapping (bwa-mem --min-read-percent-identity 95 --min-read-aligned-percent 80), the percentage of reads used in the BBayA assembly were at least > 89% for any of the *R. felis* genomes, while numbers < 72% were found for the other species...'. This is given in L451-454.

13. Table S3: I think something has gone wrong with this table. The first column shown is collection date. There are no GCF numbers, accession codes or strain names supplied in the version of the table that I have access to. This table is not interpretable without this information. Furthermore, the "collection source" column is unclear. Is "reference" supposed to indicate RefSeq? If there is a dash (-) in this column where did you get the genome from?

We thank the Reviewer for noting this. All the required information was in the previous version of the table, but it was displayed after column P. The strain name was given in column A, while the NCBI RefSeq accession number was in column C. For clarity, we searched for the origin and date of the samples in different databases to calculate the temporal signal, and therefore we obtained the collection date of the strains from different sources: NCBI, LPSN or the publication of the original samples (DOI's provided). Regarding the title of the column "collection source", it was changed by 'collection_date_source' as we obtained the collection date from different sources and in some cases, we had to choose the most confident.

14. Line 20: please specify that you are talking about human "genomic evidence" here.

This has been corrected.

15. Table 1 (page 4), what statistic is used for genomic coverage (x)? mean? Median? Did the reference you mapped to include plasmids? If so, this needs to be stated.

We corrected this in Table 1.

16. Line 127: You mention doing analysis to detect plasmid rearrangements, but you never mention it again. Did this analysis yield any results? If it was not done, it should be excluded.

We performed several analyses regarding the distribution of plasmid across the 5 *R. felis* genomes. Some of the results are given in L147-148 and L195-203. We found that the main differences between genomes was in the presence/absence of plasmids, seeming the plasmid pRF ancestral (and distributed in 4 genomes) while in the URRWXCal2 strain there was a duplication of part of this pRF plasmid and in LSU-Lb (closer to BBayA genome) an acquisition of the pLBaR plasmid through HGT. Note that the host of the LSU-Lb strain is not the cat flea (*C. felis*), but the booklouse or *Liposcelis bostrychophilus*, suggesting that this BBayA genome could have been acquired by another host besides fleas.

17. Line 128: specify whether you mapped the data individually to each reference for the 126 genomes or whether you did a competitive mapping (i.e. where you concatenate all references into one multifasta and mapped to that).

This sentence was changed regarding the comments from above. Regarding the comparison with the 126 *Rickettsia* genomes, we performed competitive as well non-competitive mapping and is clarified now in Mat/Met as said in comments above.

18. Line 156: are you referring to de-novo reconstruction here or mapping? This should be specified throughout the manuscript. Sometimes people use reconstruction to refer to mapping. And since you have done both, you should be clear when you are talking about what.

We referred to the assembly, and this is now indicated in L161.

19. Figure 3 (page 7): the branches in sub-figure C do not show up well in the PDF.

We have improved sub-figure C accordingly.

20. Line 331-332: DNA Away does not contain bleach. It contains Sodium Hydroxide which is caustic soda/lye. Therefore, the sentence: “Further handling of the specimens was done in a bleach decontaminated (DNA Away, ThermoScientific) enclosed sampling tent...” does not make sense.

This has been corrected to state that ‘Further handling of the specimens was done in a bleach-decontaminated, also using DNA-Away (Thermo Scientific) enclosed sampling tent with adherent gloves (Captair Pyramide portable isolation enclosure, Erlab)’.

21. DNA Away is referred to interchangeably as “DNA Away” (correct) and “DNA-away” throughout the methods. And “ThermoScientific” is referred to in several different ways throughout the methods. These names should be consistent throughout.

This has been corrected throughout the manuscript text.

22. Line 375: A reference (Schlebusch et al. 2017) is referred to for methods. This is not an open access paper. If methods listed in Schlebusch et al. 2017, beyond mapDamage which is described, were used in this manuscript, then those methods should be further explained in this manuscript.

We indicate that ‘The manuscript from which the data analysed in this study is derived, i.e., Schlebusch et al. 2017, is available at https://www.researchgate.net/publication/320101464_Southern_African_ancient_genomes_estimate_modern_human_divergence_to_350000-260000_years_ago’. This is now shown in L335-338.

23. Line 432-434: Repetition of the same information. Also, you need to state that the completeness and contamination stats are derived from CheckM, and not coverm which is the tool most recently referred to prior to these sentences.

We have re-structured this section to clarify the analyses performed using CheckM and CoverM, indicating that ‘The quality of all the *Rickettsia* genomes was evaluated with the CheckM v1.1.3 software package⁷³ (Table S5). Of the available NCBI RefSeq genomes used, some displayed lower completeness values (e.g., 90.97 % for GCF_000964995.1) and higher contamination values (e.g., 7.04 % for GCF_000696365.1). The completeness was 99.53%, and contamination 0.47% according to CheckM software. From the other *R. felis* strain genomes, only those from the strains

Pedreira and URRWXCal2 achieved 100% completeness, with 0% and 0.47% contamination, respectively. The remaining two *R. felis* strains, namely LSU-Lb and LSU, presented lower completeness and contamination values (at 97.47% and 1.42%, and 94.14% and 0.71%, respectively (Table S3)). Also, we indicate that 'Following this, the reads that were used to assemble the *R. felis* genome were mapped against the obtained assembly, as well the other 126 *Rickettsia* genomes, using Coverm v0.6.1 (<https://github.com/wwood/CoverM>) with the BWA-MEM mapping tool, resulting in a) a competitive mapping with the five *R. felis* genomes recruiting over 80% of reads, b) in a non-competitive mapping (bwa-mem --min-read-percent-identity 95 --min-read-aligned-percent 80), the percentage of reads used in the BBayA assembly were at least > 89% for any of the *R. felis* genomes, while numbers < 72% were found for the other species, and c) the percentage of bases of the genome covered for at least 1 read at 95% identity and 80% of read alignment, was > 98% for the *R. felis* genomes, excepting the *R. felis* str. Pedreira with 94%, GCF_000964665.1, while for other species it was < 80% (e.g., GCF_000828125.2, *R. asembonensis*). The obtained *R. felis* BBayA genome consists of 1,512,774 bases and 69 contigs, with a N50 of 42,410 bases and the longest contig comprising 121,989 bases. It exhibits a GC value of 32.5% and a coding density of 84.58% for 1,561 predicted proteins'.